# A transcriptional metastatic signature predicts survival in clear cell renal cell carcinoma

Adele M. Alchahin[1,11], Shenglin Mei [2,11] ✉, Ioanna Tsea[1], Taghreed Hirz[3,4,5], Youmna Kfoury[3,4,5], Douglas Dahl[6], Chin-Lee Wu[7], Alexander O. Subtelny[7], Shulin Wu[7], David T. Scadden[3,4,5], John H. Shin[8], Philip J. Saylor [9,11], David B. Sykes [3,4,5,11], Peter V. Kharchenko[2,4,10,11] ✉ & Ninib Baryawno [1,3,4,5,11] ✉

Clear cell renal cell carcinoma (ccRCC) is the most common type of kidney cancer in adults. When ccRCC is localized to the kidney, surgical resection or ablation of the tumor is often curative. However, in the metastatic setting, ccRCC remains a highly lethal disease. Here we use fresh patient samples that include treatment-naive primary tumor tissue, matched adjacent normal kidney tissue, as well as tumor samples collected from patients with bone metastases. Single-cell transcriptomic analysis of tumor cells from the primary tumors reveals a distinct transcriptional signature that is predictive of metastatic potential and patient survival. Analysis of supporting stromal cells within the tumor environment demonstrates vascular remodeling within the endothelial cells. An in silico cell-to-cell interaction analysis highlights the *CXCL9/ CXCL10-CXCR3* axis and the *CD70-CD27* axis as potential therapeutic targets. Our findings provide biological insights into the interplay between tumor cells and the ccRCC microenvironment.

Renal cell carcinoma is the most common renal tumor in adults, and the clear cell subtype (ccRCC) accounts for 75–85% of all cases[1]. While the localized disease can often be cured with surgical resection or thermal ablation, there is evidence of metastatic disease in approximately one-quarter of patients at the time of diagnosis[2]. Tumor cells that arise initially as a clonal expansion of transformed cells then propagate in the ecosystem of the tumor microenvironment (TME) where distinct cell populations engage in complex interactions that promote tumor growth and metastatic spread[3].

The TME is composed of stromal cells, including immune cells, endothelial cells (EC), fibroblasts, smooth muscle cells, and pericytes[4]. Biological processes within the TME, such as inflammation, hypoxia, angiogenesis, and epithelial-to-mesenchymal transition (EMT), contribute to tumor complexity and evolution and may play a critical role in promoting distant metastasis[5]. Specific components of the TME are well established as therapeutic targets. One treatment strategy in ccRCC is to inhibit the VEGF signaling that is commonly dysregulated following VHL gene inactivation within RCC

[1]Childhood Cancer Research unit, Department of Children's and Women's Health, Karolinska Institutet, Karolinska University Hospital, Stockholm, Sweden. [2]Department of Biomedical Informatics, Harvard Medical School, Boston, MA, USA. [3]Center for Regenerative Medicine, Massachusetts General Hospital, Boston, MA, USA. [4]Harvard Stem Cell Institute, Cambridge, MA, USA. [5]Department of Stem Cell and Regenerative Biology, Harvard University, Cambridge, MA, USA. [6]Department of Urology, Massachusetts General Hospital, Harvard Medical School, Boston, MA, USA. [7]Department of Pathology, Massachusetts General Hospital, Harvard Medical School, Boston, MA, USA. [8]Department of Neurosurgery, Massachusetts General Hospital, Harvard Medical School, Boston, MA, USA. [9]Massachusetts General Hospital Cancer Center, Harvard Medical School, Boston, MA, USA. [10]Present address: Altos Labs, San Diego, CA, USA. [11]These authors contributed equally: Adele M. Alchahin, Shenglin Mei, Philip J. Saylor, David B. Sykes, Peter V. Kharchenko, Ninib Baryawno. ✉e-mail: smei8@mgh.harvard.edu; peter.kharchenko@post.harvard.edu; n.baryawno@ki.se

tumor cells and would otherwise stimulate endothelial cell growth and angiogenesis[6]. Immune checkpoint blockade (ICB) targets T cells within the immune TME and has improved outcomes in patients with ccRCC both as monotherapy and in combination with other agents[6–8]. However, resistance to treatment is common and may partly be attributed to other tumor-protective roles of the microenvironment[7]. Improving therapies for metastatic RCC will require a deep understanding of how the tumor cells are specifically interacting with their microenvironment.

Comprehensive genomic studies in ccRCC have provided insights into the somatic alterations that affect tumor progression[9,10] and response to immune checkpoint blockade[11]. Single-cell RNA-seq (scRNA-seq) studies on human RCC have provided immune cell atlases based on tumor samples from treatment-naïve and treated patients. This has improved our understanding of the cell composition and cellular states of tumor-infiltrating immune cells that may contribute to the response to immunotherapy[12,13]. These studies highlight the proportion of exhausted CD8 + T cells and the function of immuno-suppressive M2-like macrophages in advanced RCC[12]. While the infil-tration of cytotoxic CD8 + T (CTLs) cells has been associated with an improved prognosis in other solid tumor types, it has been correlated with a worse prognosis in ccRCC[14]. ICB treatment in RCC can remodel the microenvironment and can modify the interplay between cancer cells and immune populations such as CD8 + T cells and macrophages, but patient responses to immune checkpoint blockade are still far from universal[15,16]. These studies have expanded our understanding of the immune components of the microenvironment though the com-position of stromal cell populations and their interactions with tumor cells remain unclear. Elucidating the specific relationship between stromal cells and tumor cells within the TME will advance our under-standing of carcinogenesis and cancer progression.

In this study, we profiled human ccRCC tumors and their matched normal control kidney tissue from treatment-naïve patients, in addi-tion to primary tumors from patients presented with bone metastases at diagnosis. This single-cell transcriptomic analysis led to the follow-ing observations: (1) tumor cells are transcriptionally similar to a sub-set of proximal tubule cells which may be an indication of the tumor cell of origin[17], (2) the synchronous comparison of primary tumor and bone-metastatic tumor tissues from two patients who presented with de novo metastases revealed a specific metastatic signature associated with poor prognosis, (3) the stromal cells within RCC tumors show the highest transcriptional difference of analyzed cell types when com-pared to adjacent normal kidney, and (4) building on other scRNA-seq studies of human RCC[12,13,17,18], we identify additional components of the TME including a cellular map of the stromal cell compartment, and of cell-to-cell interactions within the tumor that might be vulnerable to therapeutic targeting. This careful dissection of the cellular and molecular landscape of ccRCC is intended to facilitate avenues of therapeutic intervention and, ultimately, better treatments for patients suffering from ccRCC.

## Results

### Primary human ccRCC show consistent microenvironmental changes as compared to matched normal kidney tissue

To provide an overview of the molecular and cellular landscape of patient-matched normal kidney and ccRCC tissue, we performed scRNA-seq profiling (10x Chromium) from freshly resected primary ccRCC tumors (16) and adjacent normal samples from 10 patients (Fig. 1a). Nine patients were diagnosed with ccRCC and 1 patient with papillary RCC, pRCC (pRCC was excluded from analysis). Two patients (RCC-BM1-PT and RCC-BM2-PT1,2) had clinical metastases at multiple sites at the time of diagnosis (Supplementary Fig. 1a, Supplementary Data 1). Tumor tissues and adjacent normal kidney tissue collected from the same patient (from two patients, tumors in a different loca-tion on the kidney were profiled) permitted a matched comparison

and helped to control for inter-individual variation. After quality con-trol (Methods), we obtained 157,881 cells (ccRCC: 122,054 cells + normal kidney: 35,827 cells), and samples were integrated using joint analysis of the heterogenous samples (Fig. 1b and Supplemen-tary Fig. 1b).

Unsupervised clustering identified 21 distinct clusters, including normal kidney cell populations, tumor cells, and immune and non-immune stromal populations (Fig. 1b). The stromal cells included pericytes (expressing *RGS5* and *MYH11*), endothelial cells (*RAMP2* and *CD34*) and fibroblasts (*DCN* and *LUM*). Lymphoid cells included T cells (*CD3D* and *CD3E*), NK cells (*KLRD1* and *XCL1*), and B cells (*CD79* and *CD19*). The myeloid compartment consisted of macrophages (*CD68, C1QA, C1QB,* and *C1QC*), monocytes (*FCN1* and *S100A9*), and myeloid dendritic cells (*CLEC9A* and *CD1C*) (Fig. 1c).

The patient-matched adjacent kidney samples (normal, tumor uninvolved) allowed us to identify tumor-specific changes. Cell frac-tion differences were performed using cell density analysis on the joint UMAP embedding and direct comparison of cell proportions. At the global level, this analysis revealed an enrichment of pericytes in the RCC as compared to their adjacent normal kidney tissues, an increase in the CTLs and proliferating T cells, as well as macrophages (Fig. 1d, e and Supplementary Fig. 1c, d). As proportional changes of one subtype could potentially skew the representation of other subtypes, we con-firmed findings via a Compositional Data Analysis technique to esti-mate compositional changes[19] (Fig. 1f, Supplementary Fig. 1d).

In addition to the changes in the proportion of cell populations, we examined transcriptional state differences between tumor and adjacent normal kidney tissues using an expression distance mea-surement based on the Pearson linear correlation. The stromal com-partment, including fibroblasts, endothelial cells, and pericytes, demonstrated the largest transcriptional differences between tumor and adjacent normal (Fig. 1g). Specifically, there was upregulation of genes associated with cell motility and angiogenesis (blood vessel morphogenesis) (Supplementary Fig. 1e, f). This suggests cancer-specific alterations in the stromal microenvironment during tumor progression. Expression distances were further projected using mul-tidimensional scaling (MDS) (Fig. 1h), resulting in consistent separation of normal kidney tissue (circular) from primary RCC (triangular).

### ccRCC tumors establish an immunosuppressive tumor microenvironment

Several types of cancer, including RCC, are heavily infiltrated by immune cells even when localized[12,13]. Subcluster analysis of myeloid cells identified three subpopulations of myeloid dendritic cells (mDC), three populations of monocytes (Mono-1, 2, 3), and three populations of macrophages (Macro-1, 2, 3) (Fig. 2a). In the mDCs group, CD1C+ mDC (*CD1C, FCER1A,* and *CLEC10A*) (Supplementary Fig. 2a) were reduced in the tumor compartment compared to the adjacent kid-ney (Fig. 2b).

We identified an increasing population of proliferating myeloid clusters expressing both mDC and macrophage signatures (*MKI67, KIAA0101, CD1C,* and *C1QA*) (Supplementary Fig. 2c, d). Focused sub-cluster analysis of these cells revealed two clusters: proliferating macrophage (*C1QA* and *APOC1*) and proliferating mDC (*CLEC10A* and *CD1C*); the proliferating macrophages showed a similar cell phenotype to Macro-1/2 expressing *CD163, TREM2,* and *SPP1* (Supplementary Fig. 2a, b).

The three macrophage subpopulations showed a distinct gene signature (Macro-1: *SEPP1, PDK4,* and *FCGR1A*; Macro-2: *SPP1, CXCL9, CXCL10,* and *CD68*; Macro-3: *PLAUR, IL1B,* and *CXCL2*) (Supplementary Fig. 2b). Macro-1 and Macro-2 showed a significantly increased M2 macrophage signature score compared to Macro-3 (Fig. 2c) and expressed typical M2 marker genes (*CD68, TGFB1, CD163, TGFB2, CCL18, MMP14, CTSD, MARCO*[20], and *CSF1R*) (Supplementary Fig. 2a) suggesting that these macrophages are suppressive of the immune

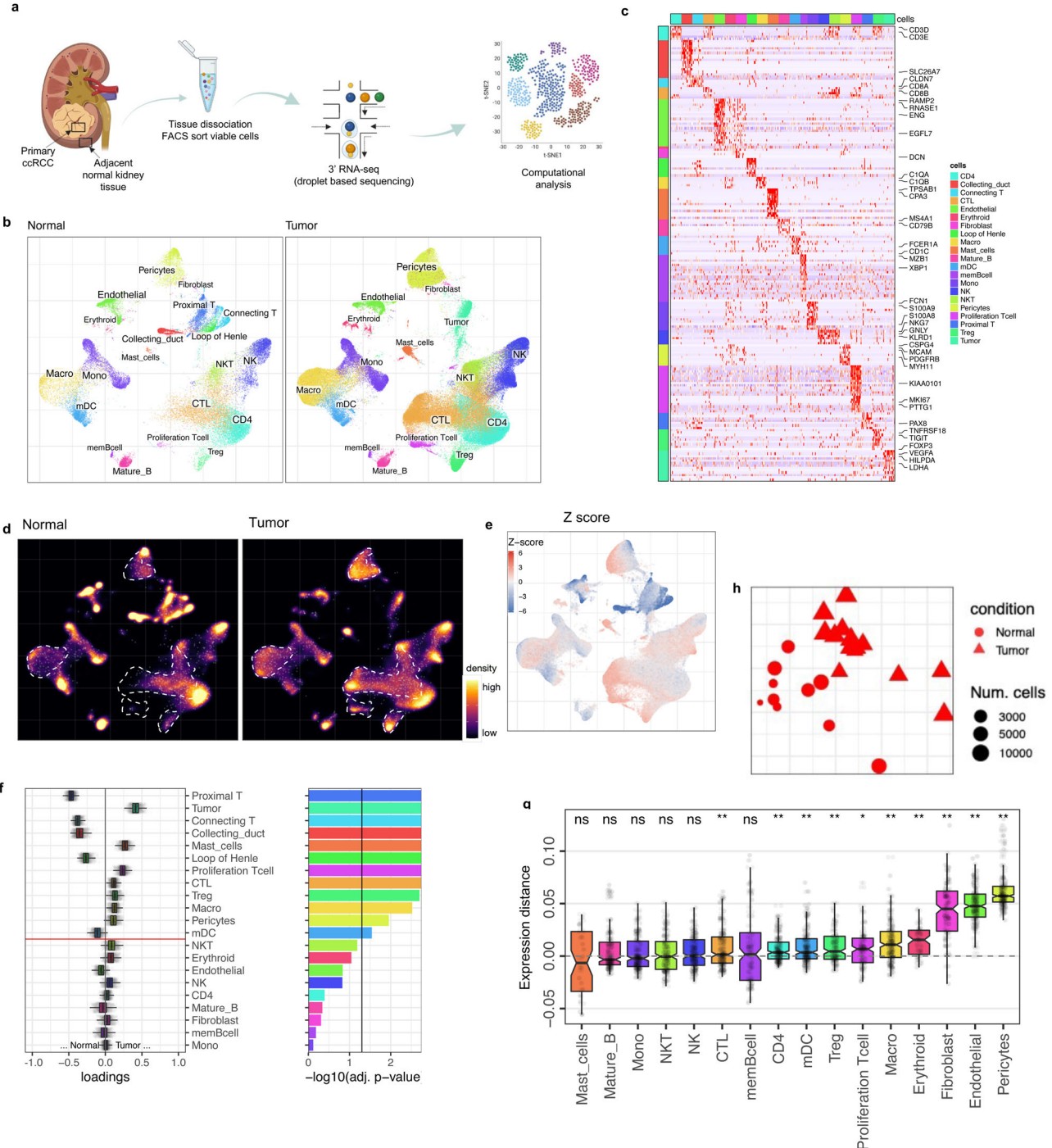

**Fig. 1 | Single-cell landscape of the ecosystem in primary ccRCC and adjacent normal tissue. a** Experimental design created with Biorender.com and Adobe Illustrator. **b** Integrative analysis of scRNA-seq samples from 26 RCC samples, visualized using a common UMAP embedding for adj-normal (left) and tumor kidney tissue (right). **c** Heatmap showing expression of markers for major cell populations. **d** Changes in the composition of all compartments combining all sample fractions and is visualized as cell density on the joint embedding. **e** Statistical assessment of the cell density differences comparing tumor with adjacent normal. A two-side Wilcoxon test was used, visualized as a Z score. Red indicates increased cell abundance in tumor, blue indicates decreased cell abundance in tumor. **f** Change in cell composition evaluated by Compositional Data Analysis. The x-axis indicates the separating coefficient for each cell type, with the positive values corresponding to increased abundance in tumor, and negative to decreased abundance. The boxplots and individual data points show uncertainty based on bootstrap resampling of samples and cells (see Methods). Boxplot includes center line: median; box limits: upper and lower quartiles;

whiskers extend at most 1.5× interquartile range past upper and lower quartiles. **g** The boxplots showing the magnitude of transcriptional change between primary RCC and normal kidney tissue in major cell populations. The magnitude is assessed based on a Pearson linear correlation coefficient, normalized by the medium variation within primary RCC and normal kidney tissue (see Methods). Statistics significance within each cell type is measured with permutation test in sample group (Pericytes \*\**p* = 0.003; Endothelial \*\**p* = 0.003; Fibroblast \*\**p* = 0.003; Erythroid \*\**p* = 0.005; Macro \*\**p* = 0.004; Proliferation T cell \**p* = 0.01; Treg \*\**p* = 0.009). Boxplot includes center line: median; box limits: upper and lower quartiles; whiskers extend at most 1.5× interquartile range past upper and lower quartiles. **h** MDS embedding of different samples, based on their overall expression distance. The similarity measure measures the magnitude of expression change for each subpopulation, using size-weighted average to combine them into an overall expression distance that controls the compositional differences. Shape indicates different sample fractions. Source data are provided as a Source Data file.

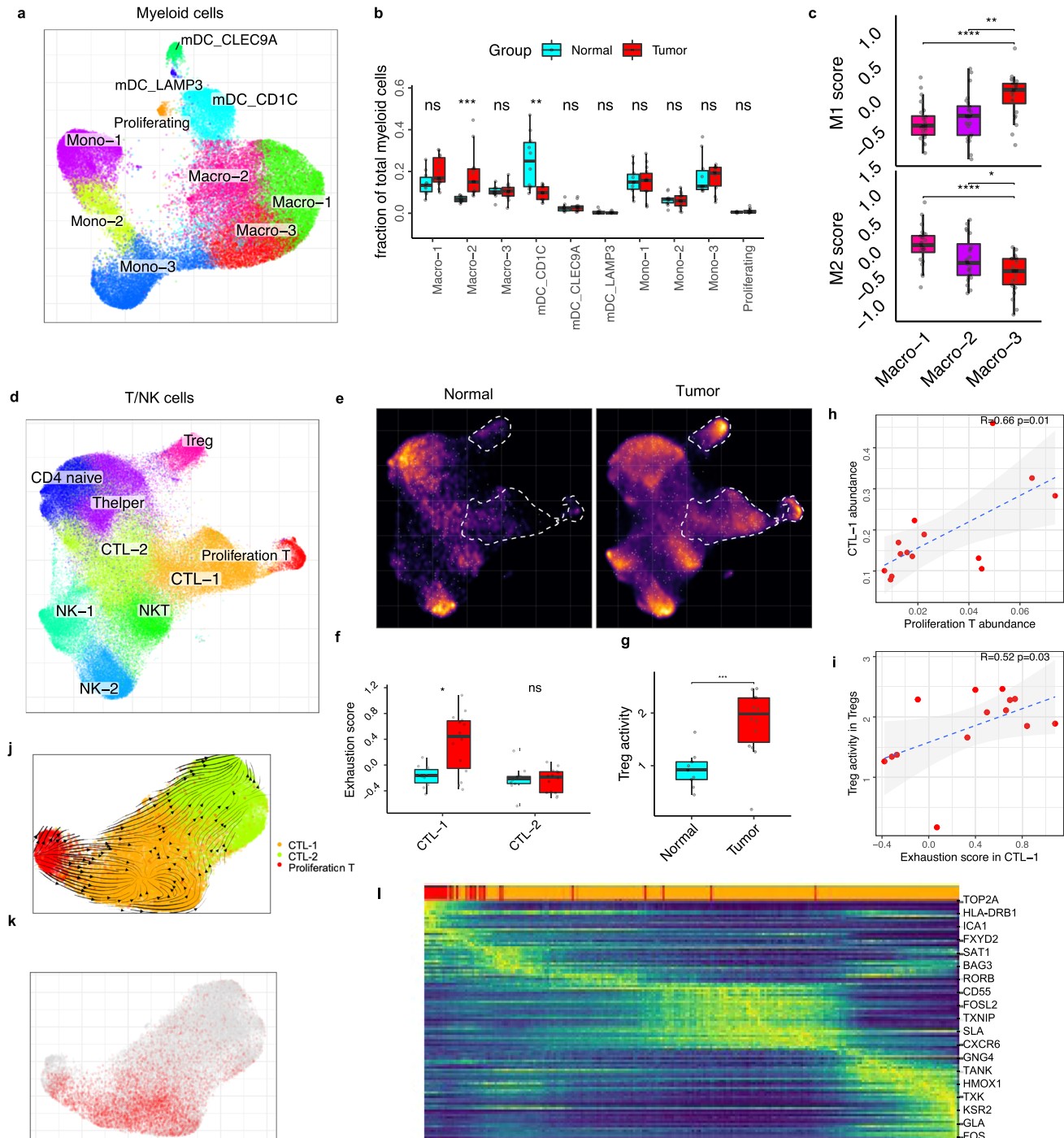

**Fig. 2 | An immunosuppressive environment in ccRCC. a** UMAP embedding demonstrating myeloid subpopulations. **b** Boxplots showing the proportions of myeloid subsets divided by the total myeloid cell number (macro-2 ****$p$ = 1.2e-05; mDC_CD1C **$p$ = 0.0024), based on two-side Wilcoxon rank-sum test (tumor $n$ = 14 samples, normal adjacent kidney, $n$ = 10 samples). **c** Average expression of the M1 and M2 macrophage signature gene panel across different monocyte populations shown as boxplot. Statistics are accessed using two-side Wilcoxon rank-sum test (M1: macro-1 vs. macro-3 ****$p$ = 3.4e-05; macro-2 vs. macro-3 **$p$ = 0.0019. M2: macro-1 vs. macro-3 ****$p$ = 4.2e-06; macro-2 vs. macro-3 *$p$ = 0.046). **d** UMAP embedding showing T-cell subpopulations. **e** Changes in the composition of the myeloid compartment between tumor and normal is visualized as cell density on the joint embedding. **f** Boxplot presenting the exhaustion score of the T-cell population comparing the adjacent normal kidney samples (turquoise) with tumor samples (red). Statistics are accessed using two-side Wilcoxon rank-sum test. IQR range similar to panel **b**. Single-cell samples, tumor

$n$ = 14 samples, normal, $n$ = 10 samples. CTL-1 *$p$ = 0.013; CTL-2 $p$ = ns. **g** Boxplots illustrate significant increase of Treg activity in the primary ccRCC. Statistics are accessed using two-side Wilcoxon rank-sum test. Treg ***$p$ = 0.00065. Boxplots in **b**, **c**)and **f**, **g** include center line, median; box limits, upper and lower quartiles; whiskers are highest and lowest values no greater than 1.5× IQR. **h** Correlation of proliferation T cells abundance and CTL-1 abundance is shown as scatter plot. Pearson linear correlation estimate, and $p$-values are shown. The error band indicates 95% confidence interval. **i** Correlation of exhaustion signature score in CTL-1 and Treg activity score in Tregs is shown as scatter plot. Pearson linear correlation estimate, and $p$-values are shown. The error band indicates 95% confidence interval. **j** RNA velocity analysis of the transitions of CTL-1, CTL-2, and proliferating T cells. **k** Visualization of exhaustion score shown on T-cell UMAP embedding. **l** Expression trends of the top 200 genes whose expression correlates with velocity pseudotime in panel **j**. Source data are provided as a Source Data file.

response and likely supporting tumor growth[13]. Macro-3 showed a significantly higher M1 signature score compared to Macro-1 and Macro-2 (Fig. 2c) and expressed high levels of *IL1A* and *IL1B*[12] (Supplementary Fig. 2a), indicative of a pro-inflammatory state. Macro-2 was separated from the other macrophage clusters by overexpressing *TREM2* and *SPP1* (Supplementary Fig. 2a, b), two genes that have been associated with tumor angiogenesis and immune checkpoint therapy[21]. The expression of *SPP1*+ tumor-associated macrophages has previously been identified in eight other tumor types, including colorectal cancer and breast cancer[21]. The overexpression of *TREM2*[22] in macrophages in tumors has been linked to resistance to immune checkpoint therapy[23]. Consistent with this, *SPP1* and *TREM2* expression were significantly increased in the tumor fraction compared to the adjacent normal kidney tissue (Supplementary Fig. 2e). *TREM2* is exclusively expressed in the macrophage population (Supplementary Fig. 2f), and further analysis of bulk RNA-seq data[24] shows that *TREM2-high* tumors are associated with poor survival outcomes (Supplementary Fig. 2j, k), suggesting that *TREM2* + M2 macrophages play an important role in ccRCC progression. Furthermore, we validated the increased *TREM2* + macrophage in tumors using flow cytometric analysis (Supplementary Fig. 2g, i) and found infiltrated *TREM2* + cells within the ccRCC tumor microenvironment by utilizing the public ccRCC spatial transcriptomic data[25] (Supplementary Fig. 2h).

Subcluster analysis of the T lymphocytes revealed the anticipated T-cell subpopulations, including CD8+ cytotoxic T lymphocytes (CTLs) (*CD8A* and *IFNG*), T$_{regs}$ (*IL2RA, CTLA-4,* and *FOXP3*), CD4+ naïve T cells (*CCR7*), T helper cells (Th) (*RORC* and *IL17A*), and subgroups of NK cells (*NKG7* and *NCR1*) (Fig. 2d; Supplementary Fig. 3a). We observed proliferating T cells (*MKI67, CD8A,* and *TOP2A*), and two different CTL populations: CTL-1 (*HAVCR2* and *PDCD1*) and CTL-2 (*CD8A* and *KLRG1*) (Fig. 2d; Supplementary Fig. 3a). In comparison to the adjacent kidney, the proportion of CTL-1 and proliferating T cells were significantly increased in the tumor fraction (Fig. 2e; Supplementary Fig. 3b). Further, CTL-1 expressed known immune-inhibitory molecules such as *PDCD1, TOX, HAVCR2, LAG3,* and *CTLA-4* indicating that the CTL-1 cells are suppressed in RCC[12] (Fig. 2f; Supplementary Fig. 3a-d). The CTL-1 exhaustion score[26] was significantly higher in tumor tissue compared to adjacent kidneys (Fig. 2f), suggesting that the tumor-associated CTL-1 might have diminished function. In parallel, we observed an increased T$_{reg}$ activity signature score in the tumor fraction (Fig. 2g), with an association with CTL-1 exhaustion (Fig. 2i; Supplementary Fig. 3f).

Within the tumor fraction, CTL-1 abundance was significantly correlated with the proliferating T-cell cluster (Fig. 2h). RNA velocity analysis can be used to infer precursor progeny cell dynamics[27], and we identified a directional flow suggesting that the proliferating T cells give rise to the CTL-1 population (Fig. 2j–l; Supplementary Fig. 3e).

The presence of NK cells in RCC may represent a critical component of the antitumor response, and NK cell infiltration in patient tumors has been associated with an improved clinical prognosis[28]. Comparing the adjacent normal kidney tissue and the tumor, we detected two subpopulations: NK-1 and NK-2 (Fig. 2d), which were further annotated as CD56$^{dim}$ NK-1 cells (*CD44, XCL1, XCL2,* and *KLRC1*) and CD56$^{bright}$ NK-2 cells (*FGFBP2, CX3CR1,* and *GZMB*)[29] (Supplementary Fig. 3a, h, i). Using a cytotoxicity score to confirm that they were functionally distinct (Supplementary Data 5), we demonstrated that the CD56$^{bright}$ subset of NK cells has a significantly greater cytotoxic phenotype (Supplementary Fig. 3i) with an expression of cytotoxic genes (*PRF1, GZMA, GZMB, GZMH,* and *GZMM*)[29] (Supplementary Fig. 3b; h, i).

## A distinct metastatic tumor cell cluster correlates with a poor prognosis

It is believed that primary ccRCC develops in the proximal part of the nephron[30], and we focused on the adjacent normal kidney tissue,

where we identified four distinct subpopulations in the nephron proximal tubule (PT), PT1-4 (Supplementary Fig. 4a, b). The PT1 transcriptional signature was characterized by metabolism-associated genes (*FABP1* and *PRODH2*)[31] and the PT4 signature of kidney fibrosis (*MMP7*[32] and *ITGB6*) (Supplementary Fig. 4b). Interestingly, in PT2 the transcriptional signature was one of inflammation and regeneration gene expression (*SOX9, IL32,* and *VCAM1*) (Supplementary Fig. 4b), which has been related to carcinogenesis and cancer progression[33,34]. Neighboring the proximal tubule cells was a small subpopulation of glomerular podocytes (*SEMA3G, CLDN5,* and *CR1*)[35] found in the Bowman's capsule, where they wrap around capillaries of the glomerulus (Supplementary Fig. 4a, b).

To relate the malignant cells to normal kidney anatomy, we compared the transcriptional states of tumor cells and normal nephron cell subsets from the adjacent normal kidneys. Tumor cells and proximal tube cells showed a similar transcriptional profile, and the malignant population clustered most closely with PT2 in the joint integration (Fig. 3a, b), pointing to PT2 as a possible tumor cell of origin. Young et al.[17] found a subset of PT cells (*VCAM1*+, *SLC17A3*+, and *SLC7A13*−) could be the origin of ccRCC, aligned with our annotated proximal tubule 2 (PT2) and PT1 (Supplementary Fig. 4c). We further utilize bulk RNA-seq data and found a significant upregulation of *SOX9, IL32,* and *SLC22A6* (PT2), and downregulation of *SLC22A6/ SLC22A8 (*PT1*)* in tumor compared to adjacent normal tissue (Supplementary Fig. 4d), confirming our cell origin hypothesis on PT2 cells.

Comparing the normal PT cells, we investigated differentially expressed genes with a focus on transcriptional programs known to underlie tumorigenesis in ccRCC. Upregulated genes included *VEGFA, NDUFA4L2, NNMT2,* and *PLIN2*, all previously shown to be involved in tumor development and progression (Fig. 3c, Supplementary Fig. 4c, d)[36,37].

We next inferred large-scale chromosomal copy number variations (CNVs) based on transcriptomic data using inferCNV[38]. Our findings are in line with previous descriptions of frequent aberrations in ccRCC, including the loss on chr3p, chr14, and gains on 5q (Fig. 3d; Supplementary Fig. 5a)[9,39]. One papillary RCC sample (PR6) displayed a markedly different CNV profile (gain of *chr3p, chr7,* and *chr17*) and was therefore removed from subsequent analysis.

Recent studies have revealed recurrent CNVs in patients with kidney cancer (TRACERx Renal and TCGA KIRC cohorts)[10,40]. To analyze tumor cell heterogeneity, we performed hierarchical clustering of inferred CNV profiles with a comparison to publicly available single-cell resolution datasets from advanced ccRCC patients[13,17]. Clustering of CNV profiles revealed four major tumor clusters that were shared by different patients (Fig. 3d; Supplementary Fig. 5c). C1 and C2 were found mainly in primary RCC patients without metastatic disease, defined by loss of chromosome 3p (Fig. 3d, e), driven by genes associated with catabolic and metabolic processes (Supplementary Fig. 5c). C3 was characterized by losses on *chr3p, chr6p, chr14,* and showed high expression of *APOE, APOC1, ACSM2A,* and *ACSM2B* (Supplementary Fig. 5d). C4 had the most abundant copy number aberrations and was notably enriched in the patients with multiple metastatic sites of disease (Fig. 3d-e). C4 also had a distinct transcriptional profile with upregulation of *SAA2, SAA1, APOL1,* and *MET* (Supplementary Fig. 5d), exclusive when compared to other cell types (Supplementary Fig. 5e). *MET* is a tyrosine kinase receptor involved in cell proliferation, survival, and migration[41]; it is among the therapeutic targets of cabozantinib[42]. We further perform DE gene analysis and defined a metastatic signature gene set (*SAA1, SAA2, APOL1,* and *MET*, Methods). Notably, we observed that the metastatic signature was significantly associated with poor prognosis in two independent RCC cohorts (Fig. 3g). Even the gene expression of the individual genes *APOL1* and *SAA1* (Fig. 3f, h) correlated with the disease stage. In addition, we also validated the metastatic signature gene expression in tumor cells from bone metastasis sites using an independent study of the bone marrow renal

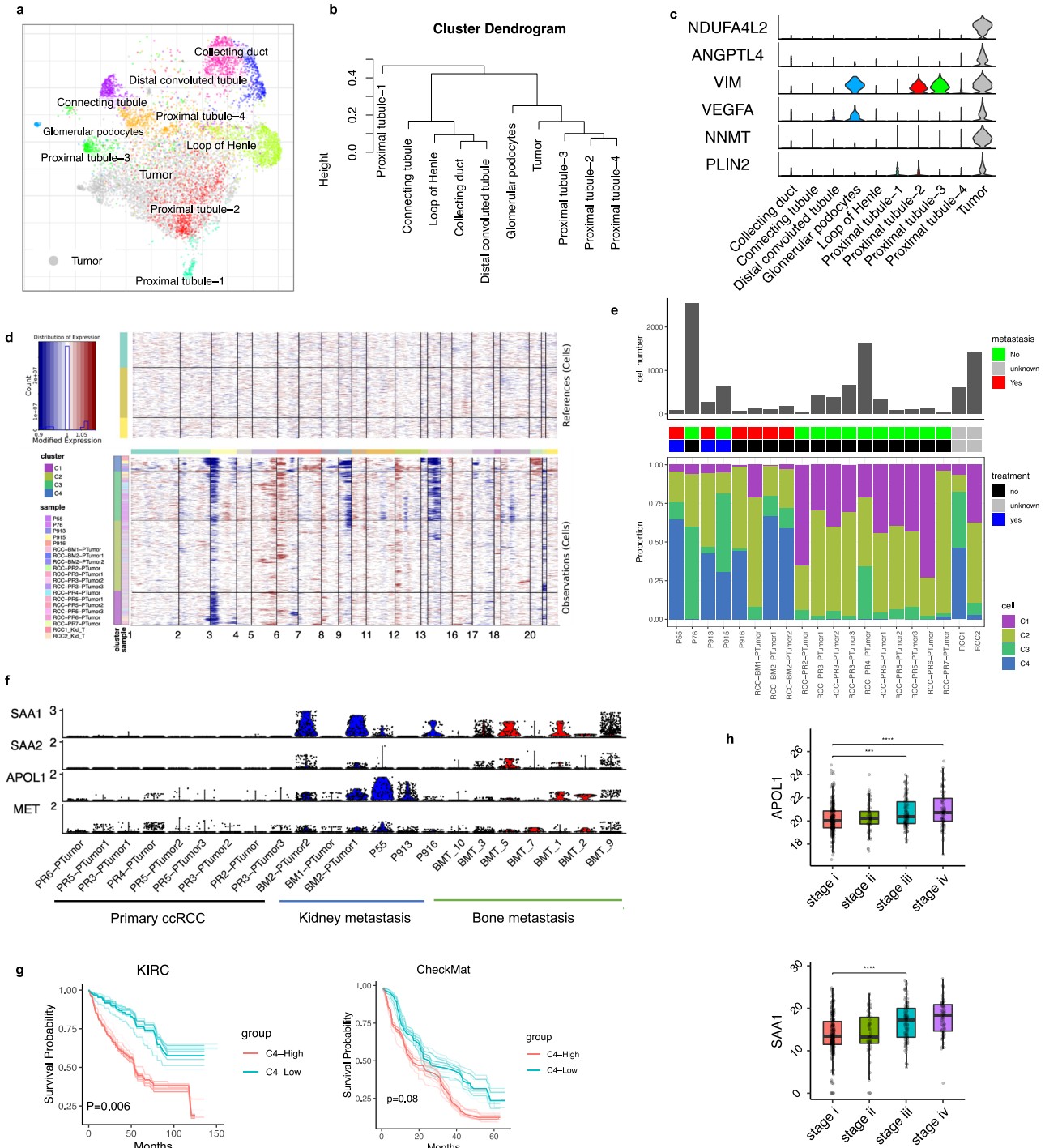

**Fig. 3 | Intratumoral heterogeneity reveals distinctive malignant cell subclones. a** Joint alignment of nephron anatomy cells from normal kidney tissue and tumor cells from primary ccRCC tissue in UMAP embedding, colored by cell annotation. **b** Expression distance of different cell subpopulations is shown as a dendrogram. **c** Violin plot showing representative marker gene expression of tumor cells. **d** Inferred CNV profile of tumor cells with normal nephron anatomy cells as normal reference. **e** Summary of tumor cell subclones, number of tumor cells (Top), clinic pathological features (middle), cell abundance of tumor subclones (bottom) in each sample. **f** Violin plot showing metastatic signature gene expression in patient tumor cells from primary ccRCC, local kidney metastasis and bone metastasis. **g** Overall survival (OS) analysis for TCGA KIRC (n = 533) and

CheckMate (n = 250) bulk RNA-seq data. Patients were stratified into two groups based on the average expression (binary: top 25% versus bottom 25%) of metastatic signatures as annotated by key marker genes in panel **f**. Statistics are accessed by two-side log-rank test. Bootstrap resampling was performed on signature genes and p-value was calculated using the 95% reproducibility power p-value (see Methods). **h** Expression of *APOL1* and *SAA1* are shown as boxplot, stratifying patients by disease stage (TCGA KIRC) with a two-side Wilcoxon rank-sum test *(APOL1:* stage i–iii ***p = 0.00016; stage i–iv ****p = 1.9e-05. Sample size: stage i = 267; stage ii = 58, stage iii = 123; stage iv = 82). Boxplots: center line: median; box limits: upper and lower quartiles; whiskers extend at most 1.5× interquartile range past upper and lower quartiles. Source data are provided as a Source Data file.

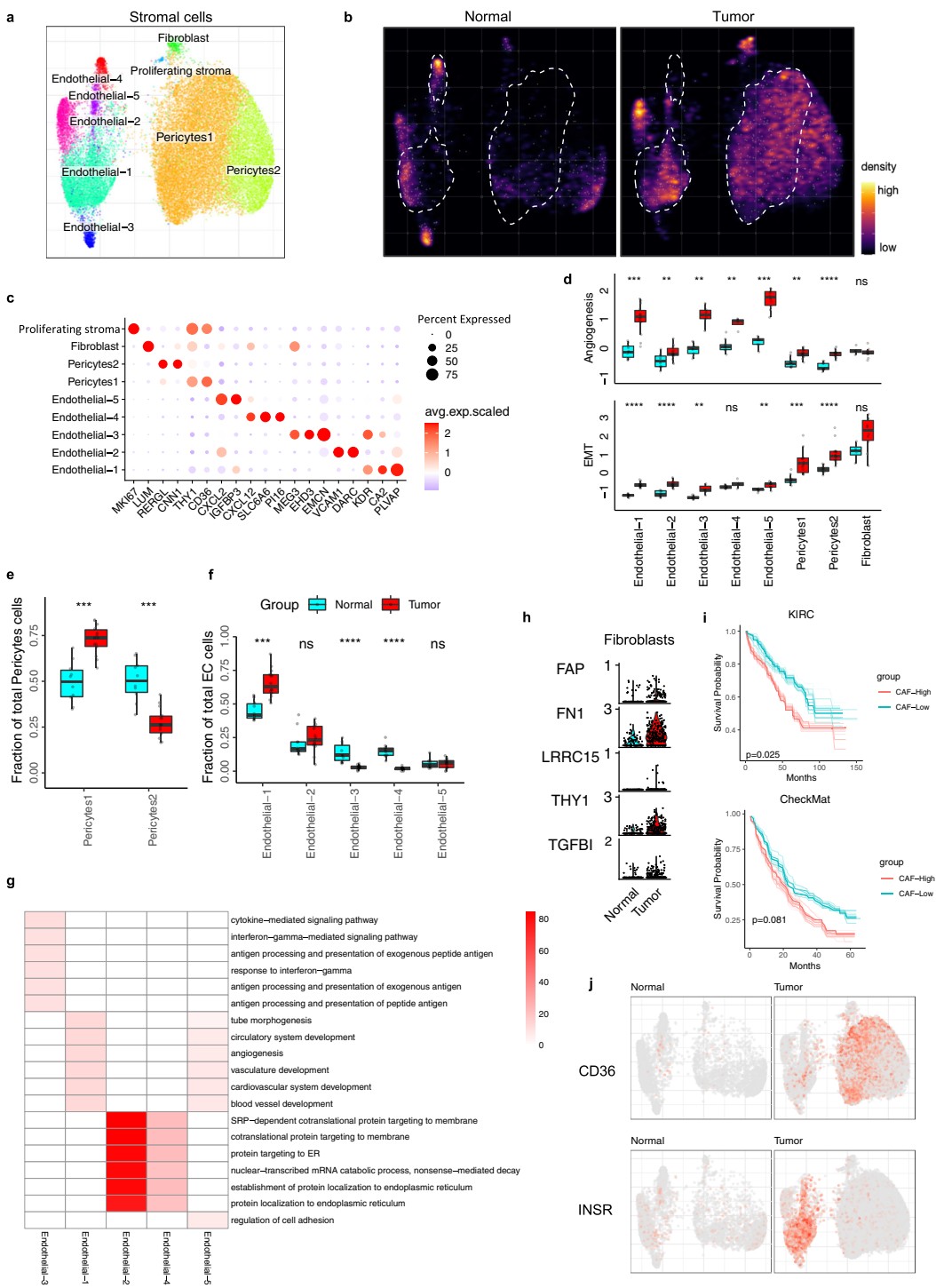

cell carcinoma (RCC) metastasis (GSE202813). Our analysis of 7 RCC metastasis cases shows the upregulation of metastatic signature compared to primary tumor cells, further demonstrating the reliability of tumor metastatic signature. Taken together, we identified a distinct tumor cell cluster with a transcriptomic program associated with metastatic potential and poor survival[43].

## ccRCC are highly enriched with vascular remodeling and EMT switch

Clear cell RCC are highly vascularized tumors with disorganized vasculature, endothelial cells, and pericytes[44]. Subcluster analysis of the stromal compartment revealed two pericytes (*RGS5, ACTA2,* and

*TAGLN*), two fibroblasts (*DCN* and *LUM*), and five endothelial (*CD34* and *VWF*) subpopulations (Fig. 4a, Supplementary Fig. 6a, b). A notable proportion of the pericytes, and some of the endothelial and fibroblast cells, were only detected in the tumor fraction (Fig. 4a, b), suggesting a tumor-induced remodeling of the stroma.

Pericyte-1 was enriched in the tumor fraction (Fig. 4e) and characterized by higher expression of genes related to a pro-inflammatory capillary phenotype (*RGS5, KCNJ8, AXL, ABCC9, PDGFRB,* and *THY1*) (Supplementary Fig. 6c). Capillary pericytes, named because they are wrapped around capillaries, favor vessel sprouting through the promotion of endothelial tip cell formation, stimulating tumor angiogenesis and possible hematogenous spread of tumor cells[45]. The

**Fig. 4 | ccRCC is driven by vascular remodeling through angiogenic and EMT switch in stromal cells. a** UMAP embedding of stromal cells, color-coded by the cell subtypes. **b** Changes in the composition of stromal cell is visualized as cell density on UMAP embedding. **c** Dot plots showing representative marker gene expression across different stromal subsets. The color represents scaled average expression of marker genes in each cell type, and the size indicates the proportion of cells expressing marker genes. **d** Boxplot plot illustrate EMT and angiogenesis signature score across different stromal cell subpopulations of normal kidney tissue and primary RCC. Statistics significances are accessed using a two-side Wilcoxon rank-sum test (Angiogenesis: endo-1 ****$p$ = 1.9e-05; endo-2 **$p$ = 0.0059; endo-3 ***$p$ = 4.0e-04; endo-4 **$p$ = 0.002; endo-5 ***$p$ = 1.00e-04; peri-1 ***$p$ = 0,00067; peri-2 **** $p$ = 1.0E-06. EMT: endo-1 ****$p$ = 1.0e-06; endo-2 ****$p$ = 1.0e-06; endo-3 **$p$ = 0.0016; endo-5 **$p$ = 0.002; peri-1 ****$p$ = 3.1e-05; peri-2 ****$p$ = 4.1e-06. tumor $n$ = 14 samples, normal adjacent kidney, $n$ = 10 samples). **e** Boxplots showing the proportions of pericytes subsets divided by the total pericytes cell number. Peri-1 ****$p$ = 1.9e-05; peri-2 ****$p$ = 1.9e-05. **f** Boxplots

showing the proportions of endothelial subsets divided by the total endothelial cell number. Statistics significances are accessed using a two-side Wilcoxon rank-sum test (Endo-1 ****$p$ = 3.1e-05; endo-3 ****$p$ = 2.0e-06; endo-4 ****$p$ = 4.7e-05). Boxplots in **d**–**f** include center line, median; box limits, upper and lower quartiles; and whiskers are highest and lowest values no greater than 1.5× interquartile range. **g** The enriched GO BP terms of top 200 upregulated genes for each stromal cell subtype comparing to adjacent normal. The statistical analysis was performed by over-representation test. **h** Violin plots showing cancer-associated fibroblast (CAF) signature gene expression in tumor and adjacent normal fibroblast. **i** Similar with Fig. 3g, showing ccRCC samples with higher CAF signature gene expression have worse overall survival in TCGA KIRC (top, $n$ = 533) and CheckMate (bottom, $n$ = 250) data. Statistics are accessed by two-side log-rank test. Bootstrap resampling was performed on signature genes and $p$-value was calculated using the 95% reproducibility power $p$-value (see Methods). **j** INSR and CD36 expression in UMAP embedding separately for tumor and adjacent normal tissue. Source data are provided as a Source Data file.

Pericyte-2 cluster exhibited a vascular smooth muscle cell (VSMC) phenotype (*ACTA2, CNN1, RERGL MYH11, TAGLN,* and *PLN*)[46] (Supplementary Fig. 6c), and the proportion of cells was significantly reduced in the tumor compartment (Fig. 4e). To validate this, we analyze spatial transcriptomic data and found Pericyte-2 markers are less present in tumor infiltrated region compared to Pericyte-1 (Supplementary Fig. 7a–c). VSMCs are thought to maintain the structural integrity of the blood vessels through their contractile properties and by inhibiting ECM degradation that would result from processes including vessel sprouting[46]. These pericyte changes may indicate that the vascular remodeling during ccRCC progression favors the pro-inflammatory and pro-invasive pericyte populations over the VSMCs.

The fibroblast cluster was annotated based on the expression of marker genes such as *DCN* and *LUM* (Fig. 4c; Supplementary Fig. 6a). Compared to adjacent normal tissue, we found cancer-associated fibroblast (CAF) signature genes[47] (*FAP, FN1, LRRC15, THY1,* and *TGFBI*) were increased in the tumor (Fig. 4h; Supplementary Fig. 6e) and associated with poor prognosis in two independent ccRCC cohorts (Fig. 4i). Interestingly, we detected a distinct stromal subcluster with a proliferating phenotype (*MKI67*), which was enriched in the tumor compartment (Fig. 4a–c, Supplementary Fig. 5e). However, the data are limited to make further conclusions of this population.

The endothelial population appears to be significantly decreased in the tumor compared to adjacent normal kidney tissue (Supplementary Figs. 1c and 7d). Within the endothelial cells, endothelial-1 (Endo-1) was the most abundant and enriched in the tumor (Fig. 4a, b, f), and characterized by an upregulation of tumor-associated genes (*PVLAP, CA2, SPARC, INSR,* and *IGFBP7*)[48] (Fig. 4c, Supplementary Fig. 6b, d). This gene signature is consistent with a tumor-associated endothelial cell (TEC) phenotype, which has been previously described and characterized by upregulated expression of pro-angiogenic factors[49]. Interestingly, a subcluster of Endo-1 was found almost exclusively in the tumor compartment (Fig. 4b), preferentially expressing known endothelial tip cell genes (*KCNE3, DLL4, EDNRB, ANGPT2,* and *SERPINE1*) (Supplementary Fig. 6b, f). These endothelial tip cells coordinate the sprouting of capillaries from pre-existent vessels and create access points for tumor cells in the blood stream[48]. In a spatial context, we also observed high expression of *PLVAP* and *CA2* within the tumor infiltrated region (Supplementary Fig. 7b, c).

Endothelial-2 (Endo-2) cells expressed genes of the vascular endothelium (*DARC, VCAM1,* and *VWF*) (Fig. 4c) and preferentially expressed venous EC genes (*GPM6A, CYP1B1,* and *MMRN1*) (Supplementary Fig. 6b, d)[48]. Moreover, upregulation of genes *CLU, NNMT,* and *S100A6*, which are known to promote RCC cell proliferation and metastasis, were found exclusively in the tumor compartment of Endo-2 (Supplementary Fig. 6b, d) and could indicate a supportive role of these cells to RCC progression[50,51].

Endothelial-3 (Endo-3) was characterized by high expression of *EHD3* (Fig. 4c), which is known to be expressed exclusively by kidney glomerular ECs (GECs)[52]. Interestingly, Endo-3 cells were abundant in normal adjacent tissue but significantly decreased in the tumor (Fig. 4f) and showed an increased angiogenesis score (Fig. 4d). GECs are crucial to the integrity of the glomerulus, and damage to these cells contributes to the progression of chronic kidney disease[52]. This cluster showed high expression of the angiogenesis-related endothelial hematopoietic gene *EMCN* and *MEG3* (Fig. 4c), which has been shown to inhibit cell inflammation in RCC[53].

The endothelial-4 (Endo-4) cluster was characterized by the expression of hematopoietic stem cell-supporting factors (*CXCL12* and *CD44*). The endothelial-5 (Endo-5) cluster showed a tumor-associated EC phenotype (*IGFBP3, ENPP2, SEMA3G, TM4SF1,* and *TIMP3*) (Fig. 4c; Supplementary Fig. 6b, d), high angiogenesis score (Fig. 4d) as well as similar upregulation of pathways related to blood vessel and circulatory development (Fig. 4g)[48].

## Ligand-receptor cell interaction analysis reveals potential therapeutic targets in human ccRCC

Although renal tumors are highly infiltrated with immune cells, RCC often successfully evades immune recognition by poorly understood mechanisms[13]. We interrogated ligand-receptor interactions with the goal of identifying potential prognostic biomarkers and therapeutic targets (Fig. 5a). Several ligands were specifically upregulated in the tumor cell population when compared to adjacent kidney, including *SPP1* and *CD70* (Fig. 5b).

*SPP1*, also known as Osteopontin (*OPN*), promotes cancer progression and metastasis through activation of NFkB signaling, which regulates extracellular matrix interactions[54]. Elevated levels of *SPP1* in ccRCC are correlated with larger tumor size, advanced stage, higher Ki-67 proliferation index, and decreased overall survival[54]. *SPP1* binds immune cells, including macrophages, NK cells, and T cells, and exerts immunomodulatory actions[55]. The *SPP1* receptor *ITGB1* was upregulated in tumor-derived NK cells, while another *SPP1* receptor *ITGA4* was upregulated in tumor-derived CTL-1 cells. This suggests that SPP1 produced by tumor cells exerts tumor-mediated immunoregulation through NK cells and T cells (Fig. 5b).

The ligand *CD70* induces apoptosis in B and T lymphocyte populations in ccRCC and has been associated with immune suppression[56]. We found that *CD70*–*CD27* signaling was upregulated in tumor tissue compared to adjacent kidneys (Fig. 5c, e; Supplementary Fig. 8b, d-g). *CD70*–*CD27* expression correlated with T-cell exhaustion (Fig. 5f). The *CD27* receptor was overexpressed in tumor-associated CTL-1 (Fig. 5b–d, g; Supplementary Fig. 8a, c), and correlated with CTL-1 exhaustion (Supplementary Fig. 9a, b). Both the ligand, *CD70*, and its receptor, *CD27*, are independently expressed in tumor tissue (Fig. 5h; Supplementary Fig. 9c). Our data suggests that

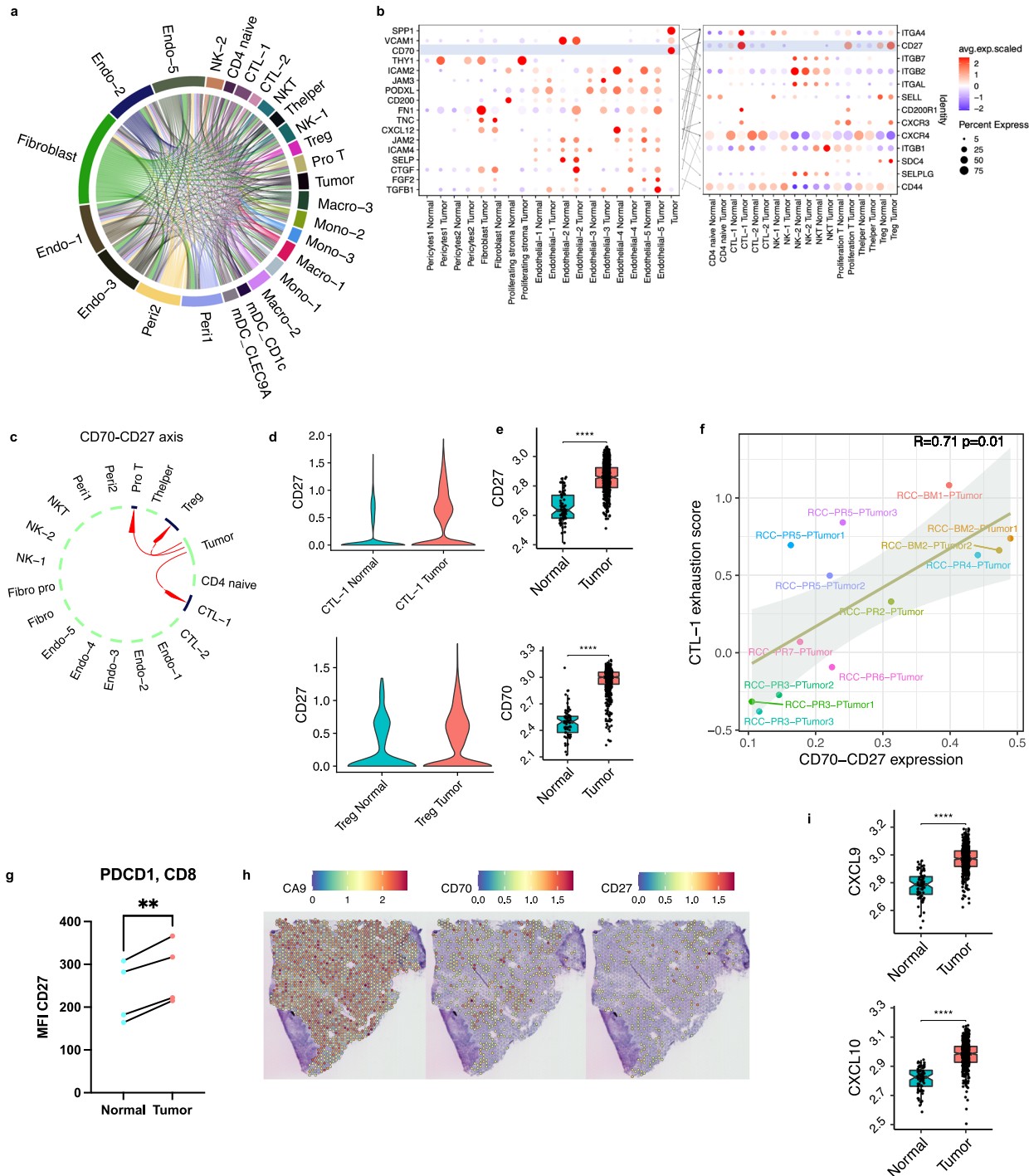

**Fig. 5 | Cell–cell interaction analysis reveals potential therapeutic targets in ccRCC. a** Overview of potential ligand-receptor interactions of cell subpopulations. **b** Bubble heatmap showing expression of ligand (tumor cells and stromal cell subsets) and receptor (immune cell subsets) pairs in different stromal and immune subsets. Dot size indicates expression ratio, color represents average gene expression. Significance of ligand-receptor pair is determined by permutation test, gene differential expression analysis and specific cellular expression (Methods). **c** The predicted interactions between *CD70* and *CD27*. **d** Violin plot showing *CD27* expression in CTL-1 and Tregs. **e** *CD27* and *CD70* expression in TCGA KIRC data are shown as boxplot. Statistics are accessed with a two-side Wilcoxon rank-sum test (*CD27* ****p < 2e-16; *CD70* ****p < 2e-16. Sample size: Tumor n = 533, Normal n = 53). Boxplots include center line, median; box limits, upper and lower quartiles; and whiskers are highest and lowest values no greater than 1.5× interquartile range.

**f** Correlation of *CD70–CD27* (tumor cells-CTL-1) average expression and CTL-1 exhaustion score is shown as scatter plot. Pearson linear correlation estimate, and p-values are shown. The error band indicates 95% confidence interval. **g** Flow cytometric analysis of *CD27* expression on *PDCD1 + CD8 +* T cells from Tumor and paired adjacent normal tissue. (n = 4 per group). Statistics are accessed with paired two-side t-test (**p = 0.0031). **h** Spatial feature plots showing *CA9*, *CD70*, and *CD27* expression in ccRCC patient. Tumor spots are marked by *CA9* expression. **i** *CXCL9* and *CXCL10* expression in TCGA KIRC data are shown as boxplot. Statistics are accessed with a two-side Wilcoxon rank-sum test (*CXCL9* ****p < 2e-16; *CXCL10* ****p < 2e-16). Sample size: Tumor n = 533, Normal n = 53). Boxplots include center line, median; box limits, upper and lower quartiles; and whiskers are highest and lowest values no greater than 1.5× interquartile range. Source data are provided as a Source Data file.

*CD70* is immunosuppressive and that its receptor *CD27* merits exploration as a therapeutic target in ccRCC[57].

The chemokines *CXCL9* and *CXCL10* and their receptor *CXCR3* (expressed on monocytes, T, and NK cells) appear to be involved in angiogenesis[58]. High expression of *CXCR3* and *CXCL9/10* has been associated with a poor prognosis, tumor growth, and increased risk of metastasis. Our analysis suggested that *CXCL9*-expressing mDC_*LAMP3* cells, as well as *CXCL10*-expressing Macro cells, interact with *CXCR3*-expressing proliferative T cells. In addition, *CXCL9*- and *CXCL10*-expressing Macro-2 tumor cells interact with *CXCR3*-expressing proliferating T cells, $T_{regs}$, and CTL-1 cells in the tumor (Fig. 5i; Supplementary Figs. 8h and 9d, g). We demonstrated overexpression of *CXCL9* in mDC_*LAMP3* and Macro-2 cells, and overexpression of *CXCL10* in proliferating macrophages (Pro Macro) and Macro-2 cells in the tumor (Supplementary Figs. 8h and 9e). We observed that *CXCL9* and *CXCL10* are correlated with CTL infiltration and *PDCD1* expression in multiple clinic cohorts from TIDE[59] (Supplementary Fig. 9g). We, therefore, hypothesize that *CXCL9/10* signaling via *CXCR3* may impact the microenvironment and potentially promote tumor progression through deregulation of inflammatory pathways.

## Discussion

Our study used coordinated surgical and research teams to bring fresh patient samples from the operating room to the laboratory. This enabled the single-cell characterization of treatment-naïve ccRCC primary tumors and adjacent normal kidney tissue with a focus on the tumor microenvironment. We uncovered intratumor heterogeneity, as well as a highly heterogenous tumor microenvironment, and multiple immunosuppressive cell–cell interactions. Further investigation of these cell–cell interactions in the tumor ecosystem highlighted potential therapeutic targets.

Clear cell RCC tumor cells almost universally display the loss of function of the von Hippel-Lindau (*VHL*) gene. Consistent with this, we observed that malignant cells have recurrent deletions of chr3p (where *VHL* is located) and upregulation of *VEGFA* and *VIM*. The cellular origin of ccRCC has been suggested to be from proximal tubule epithelium[3]. Indeed, malignant cells show similar transcription profiles with the PT2 cluster (Fig. 3a), pointing towards the PT2 subset as the ccRCC cell of origin. Intratumor heterogeneity has been widely reported[60,61], and we identified four subsets of malignant cells. Notably, malignant cell cluster 4 was associated with poor overall survival and was enriched in patients with metastases. Cluster 4 was characterized by the expression of *SAA1*, *SAA2*, and *APOL1*, genes that are upregulated in high-grade ccRCC tumors. Indeed, high *SAA1* protein expression has been negatively correlated with patient survival[43]. We suggest that Cluster 4 cells exhibit a distinct transcriptional signature that portends metastatic potential, and a specific focus on this cluster may highlight potential therapeutic targets to prevent or treat metastatic spread.

Within the complex immune microenvironment, we identified exhausted T cells (CTL-1) and immunosuppressive cell populations, including $T_{regs}$ and Macro-2. During cancer progression, CTLs can exhibit loss of function as they become exhausted due to immune-related tolerance and immunosuppression. Cancer-associated fibroblasts (CAFs), M2 macrophages, and $T_{regs}$ may counteract the CD8[+] T and NK cell-mediated antitumor immune responses[62]. Studies in melanoma have shown that increased expression of *PD-L1, IDO,* and *FoxP3* protein in $T_{reg}$ of the TME is driven by infiltration of CD8[+] T cells, further supporting the idea that these are part of an immune negative feedback loop[27]. Immunotherapies that uncouple these pathways may be the most effective on tumors showing T-cell infiltrated phenotypes. In RCC, T-cell infiltration into the TME has been demonstrated[11,12], and anti-*CTLA-4,* and anti-*PD-1* antibodies are approved treatment strategies in patients with advanced disease[63]. Both *PD-1* and *CTLA-4* receptors are upregulated in $T_{regs}$ and contribute to the immunosuppressive function of Tregs[64]. T-cell immunoglobulin mucin-3 (*TIM-3/HAVCR2*) is

another immune checkpoint surface receptor present on $T_{regs}$ that tends to suppress the immune responses and therefore resist radiation therapy[65]. This particular immunosuppressive cell population of $T_{regs}$ may represent another potential therapeutic target in ccRCC.

Regarding myeloid cells within the TME, blocking the polarization to M2 macrophages results in increased recruitment of CTLs and an antitumor immune attack[66]. Conversely, the induction of M2 macrophages impairs the response of CTLs in the TME[66]. We show that this phenomenon also exists in ccRCC, as illustrated by the increased Macro-2 (Fig. 2b) and the increased exhaustion score of the CTL-1 (Fig. 2f). The Macro-2 population expressed high levels of *TREM2* in the tumor fractions and expression of *TREM2* has been shown to be elevated in malignant tumors, including RCC[67]. *TREM2[−/−]* mice are more responsive to immune checkpoint blockade and less susceptible to cancer progression[23]. *TREM2* inhibition of the tumor-infiltrating macrophages suppressed tumor growth and enhanced checkpoint blocking therapy in preclinical studies. Here, we show that *TREM2* is elevated in the TME myeloid cells (Supplementary Fig. 2g–i), pointing to the potential as a prognostic marker in ccRCC and as a biomarker that could identify patients that would benefit from checkpoint blockade immunotherapy. Our interactome analysis suggests that the immunosuppressive role of *TREM2* macrophages (Macro-2) affects T-cell exhaustion through C-X-C motif chemokine ligand 9/10 (*CXCL9/CXCL10*) and C-X-C-chemokine receptor 3 (*CXCR3*) signaling[23].

In terms of cell–cell interactions, we identified the *CD70–CD27* relationship as potentially important, hypothesizing that upregulated *CD70–CD27* signaling in the tumor may result in CTL exhaustion. Here, the tumor cells expressed the ligands *CD70* and *CD80–CD86* that bind to their respective *CD27* and *CD28* receptors expressed on CD8[+] T cells. The receptor-ligand interactions represent the first step for CD8[+] T-cell priming[68]. When activated, naive CD4[+] and CD8[+] T cells upregulate *CD27*, resulting in increased circulating soluble *CD27*, a diagnostic marker of T-cell activation[69]. *CD70*, the ligand of *CD27*, is exclusively expressed after immune activation and is also found in both CD8[+] T cells and $T_{regs}$[69]. The ligand *CD70* is frequently overexpressed in several solid cancers, most prominently renal cancer (87%) but also including lung cancer (10%), glioblastoma (42%), and ovarian cancer (15%). Given that its receptor *CD27* is predominantly found on exhausted T cells (CTL-1) (Fig. 5g; Supplementary Figs. 8e–g and 9a, b, c), *CD70–CD27* interactions may contribute to immune evasion by the tumor. If so, these interactions would represent a potential therapeutic target[70].

In conclusion, our study provides several important biological insights about ccRCC: (1) We identify a distinct metastatic signature that predicts survival outcome, (2) we characterized a proximal tubule cell population that may represent the ccRCC cell of origin, and (3) we dissect the immunosuppressive environment and the stromal alterations in treatment-naïve patients. We highlight potential therapeutic targets to be further evaluated in preclinical models. We hope that our study will provide a valuable resource for further explorations into critical cellular relationships and signaling axes that remodel the tumor microenvironment resulting in a tumor-suppressive environment. Ultimately, validating these relationships may lead to alternative approaches to address the clinical problem of relapsed and metastatic ccRCC.

## Methods

### Experimental model and subject details

All human-subjects tissue collection was carried out with institutional review board (IRB) approval (Dana Farber/Harvard Cancer Center protocol 13–416 and Partners protocol 2017P000635/PHS).

**Tumor specimens.** In the cases, each patient was diagnosed with primary ccRCC. They underwent nephrectomy, and the patients gave consent for the use of their tissue for research purposes. At the same

time as the tumor was resected surgeon was also able to obtain a tissue adjacent to the tumor that is considered healthy normal kidney tissue for control. Specimens were submitted to pathology as standard confirmation of the diagnosis of clear cell renal cell carcinoma. The total number of patients in this analysis consists of 10 patients diagnosed with a primary ccRCC from whom we obtained the tumor material and their matched adjacent normal kidney tissue. In total 15 primary ccRCC tumors, 1 pRCC, and 10 adjacent normal kidney tissue samples were collected.

## Sample preparation

**Dissociation of tissues into single cells.** All samples were collected in Media 199 supplemented with 2% (v/v) FBS. Single-cell suspensions of the tumors were obtained by cutting the tumor into small pieces (1 mm³) in a 70 mm filter cap, followed by enzymatic dissociation for 45 min at 37 °C with shaking at 120 rpm using Collagenase I, Collagenase II, Collagenase III, Collagenase IV (all at a concentration of 1 mg/ml) and Dispase (2 mg/ml) in the presence of RNase inhibitors (RNasin (Promega) and RNase OUT (Invitrogen)). Erythrocytes were subsequently removed by ACK Lysing buffer (Quality Biological), and cells resuspended in Media 199 supplemented with 2% (v/v) FBS for further analysis.

**FACS sorting of human samples for single-cell RNA-sequencing.** Single cells from the tumor and the normal adjacent kidney samples subjected to RBC lysis were surfaces stained with anti-CD235-PE (Biolegend) for 30 min at 4 °C. Cells were washed once with 2% FBS-PBS (v/v) followed by DAPI staining (1 ug/ml). Flow sorting for live and non-erythroid cells (DAPI-neg/CD235-neg) was performed on a BD FACS Aria III equipped with a 100 um nozzle (BD Biosciences, San Jose, CA) instrument. All flow cytometry data were analyzed using FlowJo software (Treestar, San Carlos, CA).

**Massively parallel single-cell RNA-sequencing.** Single cells were encapsulated into emulsion droplets using Chromium Controller (10x Genomics). scRNA-seq libraries were constructed using Chromium Single Cell 3′ v2 Reagent Kit according to the manufacturer's protocol. Briefly, post-sorting sample volume was decreased, and cells were examined under a microscope and counted with a hemocytometer. Cells were then loaded in each channel with a target output of ~6000 cells. Reverse transcription and library preparation was done on C1000 Touch

Thermal cycler with 96-Deep Well Reaction Module (Bio-Rad). Amplified cDNA and final libraries were evaluated on an Agilent BioAnalyzer using a High Sensitivity DNA Kit (Agilent Technologies). Individual libraries were diluted to 4 nM and pooled for sequencing. Pools were sequenced with 75 cycle run kits (26 bp Read1, 8 bp Index1, and 55 bp Read2) on the NextSeq 500 Sequencing System (Illumina) to ~70–80% saturation level.

**Flow cytometric validation.** ccRCC samples and their adjacent normal kidney tissue were thawed and prepared for flow cytometry according to the pre-decided stroma, macrophage, and T-cell panels given in Supplementary Table 8. Samples were projected to FC-block, further labeled, and fixed with a live/dead assay. For stroma, we selected populations negative for CD45 from live cells, and for the immune populations, we selected CD45 positive population from live cells to proceed with the analysis. Flow cytometric and statistical analyses were made in FlowJO (version 10.8.1, Java 9.0.1 + 11) and GraphPad Prism9 (v.9.3.1 (350)).

## scRNA-seq data preprocessing and quality control

Fastq files were obtained using bcl2fastq (v2.20.0.422). The sequenced 10X libraries were then mapped to GRCh38 human genome using Cell Ranger software (version 3.0.1) with default parameters. In total, we sequenced 175,485 cells. To remove low-quality and doublets cells, we excluded cells with fewer than 700 total UMI. The obtained read count matrices were further analyzed with scrublet[71] for doublets identification. We remove cells with Scrublet scores above 0.4. After quality control, 157,881 cells from 26 samples were obtained. Detailed sample and single-cell information were listed in Supplementary Table 2 and Supplementary Table 3.

## scRNA-seq data integration and batch effect correction

We used Conos[72] (v1.4.1) (k = 15, k.self = 5, matching.method = 'mNN', metric = 'angular', space = 'PCA') to integrate multiple scRNA-seq datasets together. Principal component analysis was performed on 2000 genes with the most variable expression selected by conos. Leiden clustering was used to build to determine joint cell clusters across the entire dataset collection. UMAP embedding was estimated using embedGraph function in Conos with default parameter settings.

To identify myeloid, stroma, and T-cell subpopulations, we extracted all myeloid and all T-cell populations and realigned them separately using Conos. Leiden community detection method (as implemented in Conos) was used to determine refined joint clusters, providing higher resolution than the initial analysis.

## Determination of major cell types and cell states

To identify major cell types in both tumor sample datasets and healthy sample datasets, we used sets of well-established marker genes for each of those cell types and annotated each cell type based on highly expressed genes. We used previous single-cell atlas studies datasets for the annotation of all cell populations[18,21,73]. The detailed gene list can be found in Supplementary Data 4. For subtype assessment within the major cell types, we re-analyzed cell subsets separately with Conos. We separately extracted all myeloid cells (T cells/stromal cells) and used Conos to identify subclusters with the default settings. Each sample requires at least 40 cells, and pRCC patient was excluded from the analysis.

## Calculation of gene set signature scores

To assess cell states in different cell subsets and conditions, we use a gene set signature score to measure the relative difference of cell states. The signature scores were calculated as average expression values of the genes in a given set. Specifically, we first calculated the signature score for each cell as an average normalized (for cell size) gene expression magnitudes, then the signature score for each sample was computed as the mean across all cells. All signature gene modules were listed in the Supplementary Data 5. The statistical significance was assessed using a two-side Wilcoxon rank-sum test.

## Differentially expressed genes

For differential expression analysis between cell types, a two-side Wilcoxon rank-sum test, implemented by the getDifferentialGenes() function from Conos R was used to identify marker genes of each cell cluster. The genes were considered differentially expressed if the p-value determined Z score was greater than 3. For differential expression analysis between sample fractions (for example *Tumor* Treg vs. *adj-Normal* Treg), getPerCellTypeDE () function in Conos was utilized. As described previously, it first forms "mini-bulk" (or meta-cell) RNA-seq measurements by combining all molecules measured for each gene in each subpopulation in each sample. This results in a collection of bulk-like RNA-seq samples, and the downstream differential gene expression analysis was performed using DESeq2[74] (v 1.32.0) with default settings. A minimal number of 10 cells (of the selected cell type) were required for a sample to be included in the comparison. Differential expression gene lists for stroma cell subsets could be found Supplementary Data 6.

## Estimation of differential cell density

We use estimateDiffCellDensity function from cacoa (https://github.com/kharchenkolab/cacoa v0.3.0) to estimate differential cell density. To estimate differential cell density between tumor and adjacent normal on joint UMAP embedding, we first compute kernel density in joint embedding space for each sample using ks R package (bin = 400). Obtained density matrix was normalized by quantile normalization. The average density of each sample group was shown in Fig. 1d. To impute the differential cell density between sample groups, we performed *t*-test between sample groups in each girded bin. To avoid noise from the background, we filter bins with at least 1 cell, and the Z score is visualized as a heatmap.

## Compositional analyses

Statistical significance of proportion differences (Supplementary Fig. 1d) was evaluated using a two-side Wilcoxon rank-sum test. To avoid the potentially skew representation of other subtypes. We also performed compositional analyses (CoDA) to calculate robust estimates of compositional changes using estimate CellLoadings from cacao. In short, isometric log-ratio transformations were applied to cell type fractions, followed by canonical discriminant analyses using the candisc package to obtain weighted contrasts between cell types in tumor and normal samples. Furthermore, to account for inter-patient variability, we perform random cells to evaluate the robustness and statistical significance of the separating coefficients. In total, 1000 subsampling were performed each time evaluating 1000 randomly sampled cells from both groups. The separating coefficients are plotted (Fig. 1g).

## Expression distance analysis

Expression differences analyses were done using functions from cacoa. Expression differences between matching subpopulations were determined by first estimating "mini-bulk" (or meta-cell) RNA-seq measurements for each subpopulation in each sample. Briefly, in each dataset, the molecules from all cells belonging to a given subpopulation were summed for each gene (i.e., forgetting cellular barcodes). The distance between the resulting high-coverage RNA-seq vectors was calculated using Pearson's linear correlation on log-transformed values. To attenuate the impact of the differences in the number of cells, a total of 100 sampling rounds were performed. To obtain final normalized distance estimates (Fig. 1h), the expression distances of sample pairs between the conditions (tumor and adjacent normal) were normalized by the median expression distance of pairs within the tumor and normal conditions. To access the significance of expression distance for given cell types, we perform permutations of randomized sample groups (*Tumor* vs *adj-Normal*) labels and generated a null distribution for expression distance. Overall expression distances between samples were determined as a normalized weighted sum of correlation distances across all cell subpopulations contained in both samples, with the weight equal to the subpopulation proportion (measured as a minimal proportion that the given cell subpopulation represents among the two samples being compared). Expression distances between samples are then projected to 2D space using MSD using plotExpressionDistanceEmbedding function in cacao (Fig. 1f).

## Malignant cells subclusters

To identify the malignant cells, we performed differential expression gene analysis comparing putative tumor cells with normal kidney nephron cells. Next, we inferred large-scale chromosomal copy number variations (CNVs) with inferCNV (v1.3.3), which use a moving averaged expression profile across chromosomal intervals[75,76] compared to the normal reference. Here we perform inferCNV on putative tumor cells using the normal kidney nephron cells as the reference "normal" cells. To avoid potential biases from some samples, we take normal nephron cells from multiple patients as a reference control. We examine CNVs profile of tumor cells and define tumor subclusters using hierarchical clustering. Inferred CNV profiles of malignant cells and subclusters were shown in Fig. 3d.

## Metastatic signature identification

To identify the metastatic signature genes, we run multiple round differential expression analyses. Firstly, we performed different expression analyses of different tumor cell subclusters (C1–C4), requiring genes are highly expressed in C4. Genes are ranked by *p*-value determined Z score (top 100 genes). Next, we run differential expressed genes comparing C4 with all other cell populations (stroma, myeloid, and T cells), requiring genes specifically expressed in tumor cells (top 100 genes). Lastly, we collected tumor cells from ccRCC bone metastasis (GSE202813) sites and manfully evaluated the gene expression in primary tumor cells, and local metastatic tumor cells. After that, we define four metastatic gene signatures from tumor cells.

## RNA velocity-based cell fate tracing

To perform the RNA velocity analysis, the spliced reads and unspliced reads were recounted by the velocyto (v0.17) python package[77] based on previously aligned bam files of scRNA-seq data. The calculation of RNA velocity on low diminutions UMAP embedding was done by following the scvelo python pipeline[27]. scVelo (v0.2.3) was applied to both merged datasets and individual sample fractions. we use the dynamical model to learn the full transcriptional dynamics of splicing kinetic with the default setting. Different samples are analyzed under the same parameters, and there is no artificial adjustment of DEG.

## Regressing cell-cycle genes

We use Seurat (v4.0.6)[78] to regress out cell-cycle genes in join analysis of proliferating T cells and CTLs. First, we assigned each cell a score, based on its expression of G2/M and S phase markers with the CellCycleScoring function. Then we applied the ScaleData function to regress out cell-cycle genes. The scaled residuals of this model represent a 'corrected' expression matrix, that can be used downstream for dimensionality reduction. UMAP embedding was used to visualize CTL subpopulations.

## Spatial transcriptome data analysis

10X Visium Spatial Transcriptomics (ST) data were downloaded from GSE175540. The gene-spot matrices generated after ST data processing from ST and Visium samples were analyzed with the Seurat package. Normalization across spots was performed with the LogVMR function. Dimensionality reduction and clustering were performed with independent component analysis (PCA) using the top 30 PCs. Spatial feature expression plots were generated with the SpatialFeaturePlot function in Seurat. We use CA9 expression to separate tumor cells in infiltrated and non-infiltrated regions. R ks package was used to distinguish cells within or outside of tumor cells infiltrated region.

## Ligand and receptor analysis

Ligand-receptor interactions were inferred using a similar approach as previously described[79]. We collected 1263 well-annotated ligand and receptor pairs from cellDBphone. We first screened each of the ligands and receptors based on their expression within each cell type, requiring that the gene is expressed in more than 10% of the cells. Next, we calculated the average expression of ligand and receptor pairs across cell type pairs in normalized scRNA-seq data. The product of average ligand expression in cell type A and the average receptor expression in cell type B was used to measure LR pair expression. To evaluate the robustness and statistical significance of LR pairs, we calculated a null distribution for average ligand-receptor by shuffling cell identities in the aggregated data and recalculating ligand-receptor average pair expression across 1000 permutations of randomized cell identities. The *P*-value was the number of randomized pairs exceeding

the observed data. Benjamini-Hochberg (BH) method was used to adjust the *p*-value (<0.05). To prioritize functional ligand-receptor interaction pairs, we measured both ligand and receptor expression levels across cell types, requiring both ligand and receptor highly expressed in corresponding cell types taking other cell types as background. The LR pairs were filtered by the p-value determined Z score 3. We further performed a differential gene expression analysis between sample fractions (for example, Tumor Treg vs. Normal Treg), requiring both ligand or receptor are upregulated in the tumor fraction (log2Foldchange > 0). The detailed LR list can be found in Supplementary Data 7.

## Survival analysis
To test if a given signature predicts survival, we first computed the average gene expression level of the signature in each tumor based on the bulk RNA-sequencing data. The samples were grouped into high (top 25%) and low (bottom 25%) groups based on the average signature gene expression. Next, we used two-side log-rank test to compute the significance of the association between the signature expression value and prognosis. To evaluate the stability of the signature genes list, we resample the signature genes and repeat the analysis with 100 bootstrap resampling rounds. Statistical significance was then assessed by *p*-values at the 95% reproducibility power (i.e., reporting 0.95th quantile of the sampled *p*-values).

## Gene ontology
To test for enriched GO Biological Processes or KEGG Pathways in gene sets, the GOstat (v.4.2) R package[80] was used, with default parameters. To determine the enriched Biological Processes GO Terms, we used the approach above on the top 200 upregulated genes called by getDifferentialGenes() and getPerCellTypeDE() function in Conos.

## Statistical analysis
*P*-values < 0.05 were considered significant. A two-side Wilcoxon rank-sum test was used to assess significance in bulk seq and scRNA-seq analyses unless otherwise stated.

## Reporting summary
Further information on research design is available in the Nature Research Reporting Summary linked to this article.

## Data availability
Raw single-cell RNA-sequencing data and processed data are publicly available and can be accessed from the NCBI Gene Expression Omnibus database under accession code GSE178481. GRCh38 human reference genome was download from 10X genomics [https://support.10xgenomics.com/single-cell-gene-expression/software/downloads/]. For the joint alignment analysis with public ccRCC scRNA-seq data, we downloaded raw count matrix and cell annotation from European Genome-phenome Archive under controlled access: EGAS00001002171, EGAS00001002486, EGAS00001002325, and Single Cell Portal [https://singlecell.broadinstitute.org/single_cell/study/SCP1288/][11,13]. 10X Visium Spatial Transcriptomics (ST) data were download from GSE175540[25]. For bulk RNA-seq data, TCGA clear cell renal cell carcinoma (KIRC) cohort were downloaded from the cBioPortal [https://www.cbioportal.org/study/clinicalData?id=kirc_tcga]. The Checkmate 025 cohort data were obtained from the supplementary table of published results[11] (https://doi.org/10.1038/s41591-020-0839-y). Source data are provided with this paper.

## Code availability
The codes generated during this study are available at Github repository [https://github.com/shenglinmei/ccRCC.analysis/] and Zenodo [https://zenodo.org/record/7061983][81].

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

## Acknowledgements

We are particularly indebted to our patients and their clinical care teams. We gratefully acknowledge support from Bill & Cheryl Swanson, Gunther & Maggie Buerman, and Robert Higginbotham. A.A. and N.B. were funded by the Swedish Cancer Society. P.V.K. was funded by the NIH grant R01HL131768 from NHLBI. Patient samples were sorted at the HSCI/CRM flow cytometry core facility at MGH with the help of Maris Handley and Pathik Sen.

## Author contributions

Conceptualization: A.A., S.M., P.S., D.B.S., P.V.K., and N.B.; Investigation: A.A., S.M., I.T., T.H., Y.K., P.S., D.B.S., C.W., A.S., S.W., P.V.K., and N.B.; Computational investigation and analysis: A.A., S.M., I.T., P.V.K., and N.B.; Writing—original draft: A.A., S.M., I.T., and N.B.; Writing—review and editing: A.A., S.M., C.W., A.S., S.W., P.S., D.B.S., P.V.K., and N.B.; Funding acquisition, resources, and supervision, D.D., D.T.S., J.S., P.S., D.B.S., P.V.K., and N.B.

## Funding

## Competing interests

P.V.K. serves on the Scientific Advisory Board to Celsius Therapeutics Inc. and Biomage Inc. D.T.S. is a director and shareholder for Agios Therapeutics and Editas Medicines; a founder, director, shareholder, and scientific advisory board member for Magenta Therapeutics and Life-Vault Bio, a shareholder and founder of Fate Therapeutics, and a director, founder, and shareholder for Clear Creek Bio, a consultant for FOG Pharma and VCanBio, and a recipient of sponsored research funding from Novartis. D.B.S. is a founder, consultant, and shareholder for Clear Creek Bio. The remaining authors declare no competing interests.
