## [Peer Review File · Nature Communications]

REVIEWER COMMENTS

Reviewer #1 (Remarks to the Author): expert in clear cell RCC

Alchanin et perform scRNA analysis on a cohort of RCC patients to develop gene base biomarkers. The study adds to the growing list of studies in this space and identifies several possible therapeutic targets. Unfortunately the paper also suffers from several critical limitations:

Methodological:

Unfortunately there's no mention at all about handling the possible batch effect issue in this study. At least they should demonstrate there's no/minimal concern of batch effect in sample pooling, for example by PCA analysis. Although they, look to me, mainly used Conos (<https://github.com/kharchenkolab/conos>) for cell clustering and sub-clustering analysis which seems to have measures in integrating data from difference samples for clustering. Nevertheless there's tons of results after clustering from like, for example DEG, pseudo time, velocity and CNV analyses which could be affected by the batch effect. It's a fundamental question that how they make sure batch is not the source of reported results. For example, when comparing the expression difference between tumor cells and normal cells (or mets vs no-mets) from the same cluster, how did they make sure the difference actually came from batch effect? For CN analysis, since the read count in a sliding window basis of all the "tumor" cells from all the samples were compared to normal nephron cells, again, from all the samples, how can we be sure the normal nephron reference cells are not biased by some samples in read count? And vice versa for the tumor cell read count. For velocity analysis, the splicing event classification I believe is individual cell basis so it's not affected by the batch effect. However, when they calculated the velocity in a group of cells, they seemed to do it on both merged datasets as well as individual sample fraction but it's not clear how the bath effect was handled in analyzing merged datasets. In the velocity pseudo time analysis, since it's based on selected number of genes which often came from some kind of DEG which again could be biased by batch effect.

Another point regarding the batch effect is about signature score calculation in scRNAseq. Although it's mentioned in method that they calculated signature score for each cell as an average normalized (for cell size) gene expression magnitudes, it's not clear to me what measures were taken in avoiding score inflation or deflation by batch? Regarding the survival analysis results, there are a number of questions. First, the p values they reported are actually one-side log-rank test p value as mentioned in the method. When there's no reasonable prior assumption of the direction of the survival difference between groups, 2-side p value should be reported. For example, for the C4 signature (fig 3) survival analysis were they really testing the assumption of C4 high has worst survival as compared to low? Same questions goes to CAF, TREM2 and CTL scores. They only include the top and bottom 20% classified as high and low groups. In this case, to me this undermines the significance level and the robustness of the real biological association between the signature and the survival. They used KIRC and Braun at al cohorts for survival analysis but it's not clear which survival (OS or PFS) and arm were used in the analysis. It will be more informative to see the survival analysis results being tested in the other IO cohorts as well as recurrence cohorts if any.

Clinical: Table 1 is missing several clinical features such as tumor grade, size gender etc.

PRCC case should be excluded as its genomically and immunologically completely distinct.

ITH analysis is not addressing Turajlic point of inter-tumor heterogeneity, its more about across tumor heterogeneity. This analysis should be removed.

The authors collected bone mets on two of the patients but do not show any analysis comparing primary vs mets or any bone tropic signatures.

Reviewer #2 (Remarks to the Author): expert in immunogenomics

Summary

In this study, Alchahin et al. analyse single-cell RNA transcriptomes of samples derived from 10 treatment naïve renal cell carcinoma (RCC) patients, including biopsies from the tumour core, metastases, and adjacent normal tissue, generating data on 35,000 normal kidney cells and 122,000 tumour cells. They begin by identifying several immune and stromal cell types that are differentially abundant in tumour versus adjacent normal tissue, and transcriptional state differences within these cell types, which are pronounced in myeloid cells, cytotoxic cells, fibroblasts, endothelial cells, and pericytes. These observations set the stage for subsequent sub-cluster analysis. Alchahin et al. describe a population of TREM2⁺ SPP1⁺ macrophages that may associate with outcome and identify an 'exhausted' cytotoxic T lymphocyte (CTL) cluster which associates with tumour Treg activity. By inferring copy-number variation from single cell transcriptomes, the authors identify a cluster of cells which appear in patients who develop metastases. Differentially expressed genes in this cluster were labelled as a 'metastatic signature' which associates with poor survival and disease stage. Within stromal cells, endothelial cells and pericytes from the tumour fraction were enriched in genes associated with angiogenesis, indicating a role in vascular remodelling. Similarly, cancer associated fibroblasts (CAFs) highly expressed genes related to epithelial-mesenchymal transition (EMT), which relate to survival. Finally, from ligand-receptor pair analysis, the authors identify CD70-CD27 and CXCL9/10-CXCR3 signalling axes that may be important in the tumour microenvironment.

Overall, there are strengths in the sampling strategy, that includes both adjacent normal kidney and some metastases (although the latter limited in number). However, while there is adequate quantitative analysis characterising the transcriptional differences between tumour and normal tissue, there are several short-comings which limit enthusiasm, principally a failure to validate single cell RNAseq findings at a protein level, but also there are some analytical weaknesses, a rudimentary description of methods and this work does re-capitulate previous single cell studies in terms of inferring a proximal tubular origin of RCC.

Major comments

1. Results fig 1 – pericyte cluster is very substantial, even in normal tissue. Are the authors sure these are pericytes? What is the expression of canonical pericyte markers such as CD31, NG2, PDGFR beta, CD146, Nestin? RGS5 expression is not specific to pericytes, similarly MYH11 is expressed in other cell types, such as smooth muscle cells.

2. Results fig 1 – in a similar vein, the marker genes stated for monocytes are non-specific. Both FCN1 and S100A8/9 are highly expressed in neutrophils. Additional information is needed on the expression of monocyte and neutrophil-specific genes – eg, FCGR3B for neutrophils.
3. Results fig 1 – proportional changes in cell types. The conclusions about relative enrichment of different cell types is made based on representation in scRNAseq data. This is problematic, as different cell types are preferentially preserved during the generation of 10x data, and this is not uniform between tissues, for example, a more hypoxic tissue like a tumour will see more cell attrition during processing and loading. Any statements on abundance of cell types between tumour and normal needs to be validated by an alternative modality, eg, flow cytometry.
4. Results 2, referencing fig 2b. see comment 4 re: making conclusions about whether a cell type is less abundant in tumour. This would need to be confirmed by an alternative methods like FACS or imaging. Furthermore, the comparing the reduction of tumour-resident CD1C+ mDCs to circulating populations is not necessarily relevant as the source they may arise from precursors or monocytes rather than enter the tumour as differentiated cDCs. It should also be made clear that reference 27 is a study on blood immune populations in breast cancer, not RCC.
5. Results 2, macrophage populations. The fact that there is differential abundance of Macro-2 in tumours, which have a higher M2 score is not sufficient to merit the claim that they are 'suppressive and support tumour growth'. Are there any gene expression patterns to support a directly suppressive role for these cells, eg, ARG1, PDL1/2? Additionally, expression of CD163 and CSF1R (which the authors identify as M2 marker genes) are similar in Macro-2 and Macro-3. Furthermore, CSF1R is not an M2s-specific macrophage gene.
6. Results 2, macrophage populations. M1 versus M2 gene signature – it is not evident (I cant find this info in the methods) what genes were included in this M1 and M2 scores. Were these genesets derived from mouse or human macrophages?
7. Results 2, SPP1+ TREM2+ Macro-2 population. Firstly, validation of the presence of the TREM2 macrophage population in tumour v normal is required. Secondly, if validated that TREM2-expressing macrophages are enriched in the tumour fraction, scoring TREM-2 gene expression in bulk RNAseq data (Ext Data 2g) and concluding that TREM2+ macrophages play a role in ccRCC progression is problematic. The authors do not show that the differential expression of TREM2+ in the bulk RNAseq can be solely attributed to macrophages. We suggest that, at a minimum, the authors show data on TREM2 expression in other tumour cell populations (both immune and non-immune), and perform deconvolution of the bulk data, to make claims on the role of this population on RCC outcome. Moreover, this statistical approach is not robust. If one were to take 30,000 genes and scored each one of these in a Kaplan-Meier analysis with LogRank p-value cut-off of 0.05, then over 1000 genes would come up as 'significant' for association with survival. In addition, for all KM analyses, the definitions of 'high' and 'low' for TCGA/CheckMate cohort stratification should be more clearly defined.
8. Results 2, Ext data 3a. The plot inserted here is incorrect and is a repeat of Fig 2b.
9. Results 2, CTL-1 exhaustion score. What does this CTL-1 exhaustion score consist of? reference 33 is a systematic review with no specific reference to the gene set used for scoring. If there is an overlap between genes used to score exhaustion and Treg activity (which is also not described), then the analysis in Fig 2h and Ext. data 3f would be fundamentally flawed. Moreover, the authors do not indicate how CTL-1 exhaustion is scored, and if this were done by simple module scoring or gene set enrichment, then it is incorrect to conclude that this CTL-1 population associates with survival (Ext. data 3g), rather it is the genes in this gene set that do.
10. Results section 2, Fig 2i. Similar comment to 6). How was 'Treg activity' scored? Without showing that genes in this signature are not differentially expressed outside the Treg compartment or without performing deconvolution, the effect on survival cannot be attributed to Tregs.
11. Results section 2, RNA velocity analysis. The use of RNA velocity analysis to infer differentiation trajectories in the current format is problematic for several reasons. 1) The fact that CTL-1 and proliferating T cells are differentially abundant in the tumour is not sufficient grounds to isolate these 2 clusters only for RNA velocity analysis. If so, there are only 2 possibilities for the root/parent cluster identified, as is inferred by the authors, which is not how RNA velocity should

be used. 2) Further, if it is true that the red cluster represents proliferating T cells and the yellow cluster exhausted T cells, given that RNA velocity measures unspliced-to-spliced transcripts, then this will likely bias the analysis to select proliferating cells as the root, given they are more transcriptionally active. Indeed, RNA velocity analysis on data where there are clusters with significantly higher levels of cell proliferation often results in large arrows originating from the proliferating cluster. 3) Visualisation – the RNA velocity analysis should not be performed on a sub-setted graph derived from Fig 2d. A new graph and new embeddings should be computed for CTL-1 and proliferating T cells, with RNA-velocity overlaid. We suggest that subset selection be expanded to all T cells, or CTLs only. In the latter, it may be necessary to show that the proliferating T cells are related to CTLs only. It may be worth considering whether these are stem-like (TCF1+/TCF7+) T cells. Importantly, the conclusion that proliferating T cells give rise to the exhausted CTL-1 population must be toned down.

12. Results section 3, inferred CNV profiles and metastatic signature. There is insufficient evidence here to argue that genes associated with cluster 4 represent a metastatic signature. 3 out of 4 primary tumour samples which had corresponding metastases, and metastasis samples from a single patient had cells belonging to this cluster. The cell number from each of these samples is less than 250. While, there may be an association here, the claim that differential genes of this cluster represent 'a distinct signature predictive of metastatic potential', based solely on qualitative observations, is over-reaching.

13. Results section 3, C4 signature. The authors show that 3 out of 4 genes in their 'C4 signature' are more highly expressed in late-stage disease (Fig 3h, Ext. data 4f). They then show that this same signature also associates with poor survival (Fig 3g). This means that the survival analysis in 3g may simply reflect disease stage, which is the definition of poor outcome. Hence, the claim made in the manuscript's conclusion, stating 'We identify a distinct metastatic signature that predicts survival outcome', while not necessarily incorrect, has important caveats to disclose, in conjunction with point 12).

14. Results section 4, stromal cell populations. In this section, several claims are made regarding very rare cell populations (eg. Endothelial-3 or Fibroblasts proliferating). The authors should demonstrate that more than a single patient / sample contribute to these rare populations to prove that these are not merely a sequencing batch effect or poor integration. If this is true, then they need to validate the presence of these cells using an alternative technology to single cell RNAseq, eg, spatial method.

15. Results section 4 – claims like 'a notable proportion of the pericytes, and some of the endothelial and the fibroblast cells, were only detected in the tumor fraction (Fig. 4a, b),' need to be validated, confirming the presence of the cell types claimed to be tumour-unique or enriched in tumour and absence in normal, for example by using flow cytometry employing marker genes for these clusters identified by sc RNAseq, or by imaging, eg, RNAscope or immunofluorescence microscopy.

16. Results section 4, 'tumor fibroblasts exhibit the highest EMT gene signature, further supporting their role in the promotion of RCC progression and metastasis'. The latter part of this statement is not supported by the data shown in Fig 4d where the EMT signature is not significantly different in tumour or normal fibroblasts.

17. Results section 4, Fig 4g. While GO terms may be useful in reducing dimensions when considering large numbers of genes, they tend to be unspecific, especially given that there are 7500 GO BP terms, which may erroneously detect significant pathways. In addition to this figure, it would be more convincing to demonstrate the genes driving this association, for example, by identifying key genes driving the association in endothelial cells, or gene-set enrichment analysis in a 'pseudo-bulk' manner.

18. Results section 4, Endo-2 Fig 4g, 'Top upregulated pathways for Endo-2 include protein localization to the ER, which could indicate an increase in demand for protein synthesis and secretion'. Following on from point 18), protein trafficking and processing genes are pathways that appear repeatedly when applying GO term analysis to multiple data sets in a highly unspecific manner. Thus, the conclusion regarding Endo-2 here based solely on a single GO term is speculation at best.

19. Results section 4, Fig 4i. Similar comments on previous KM analyses. What are the genes used for scoring? What are the cut-offs used for cohort stratification? The analysis only shows that the genes are relevant rather than CAFs.

20. Results, endothelial cell clusters. Moreover, the 3 paragraphs on Endothelial cell clusters 1-5, spanning over 1.5 pages, is highly descriptive and simply restates what is already shown in figure 4 and Ext. data 5, and is speculative.

21. Results section 5 – predicted interactions - to conclude that CD70 represents a therapeutic target in RCC the authors at least need to validate expression of CD70 and CD27 at a protein level on the cell types they claim are of relevance – eg – protein level expression of CD27 on tumor-associated CTL-1, proliferating T-cells and Tregs.

22. Results section 5, figure 5f. From this data, 1 of 3 conclusions can be drawn: 1) CD27 expression is associated with CTL-1 exhaustion; 2) CD70 expression is associated with CTL-1 exhaustion; 3) CD27 and CD70 in combination is associated with CTL-1 exhaustion. To make the conclusions that the authors are claiming, i.e. the CD70-CD27 interaction is important in the TME, 1) and 2) have to be ruled out.

23. Results section 5, figure 5j. It is unclear what this figure is trying to show, and the legend associated is not informative and may potentially have an error?

24. Discussion, 'unexpected intratumor heterogeneity'. The reviewers do not believe that the manuscript has uncovered 'unexpected' heterogeneity, as it is largely as previously described in scRNAseq datasets.

25. Discussion, 'an increase in Tregs expressing HAVCR2 is accompanied by increase of exhaustion of CTL-1 in the tumor fraction. This particular immunosuppressive cell population of Tregs may represent another potential therapeutic target in ccRCC.' The authors suggest that these Tregs represent an additional therapeutic target independent of the PD-1 and CTLA-4 axis. But, are these TIM-3+ Tregs different from the PD-1 and CTLA-4 expressing Tregs? This is not addressed.

Minor comments

1. Introduction, paragraph 3 – typo '(several?)' appears to be an edit that should be deleted

2. Introduction, final paragraph. '...tumor cells are transcriptionally similar to a subset of proximal tubule cells which may be an indication of the tumor cell of origin²³,' – in correct reference – The correct reference here is Young, Mitchell, Braga et al. Science (2019) (reference 25). This paper presented one of the earliest and largest scRNAseq studies in RCC, and is the key publication to cite when describing transcriptional similarity between the RCC tumour cell of origin and PT, and when citing RCC scRNAseq papers. This reference should similarly be included in later results and discussion sections discussing this point.

1. Figure 1h. This MDS plot highlighting different expression distances between tumour and normal samples adds little value and is unsurprising. Moreover, visualising tumour and normal by shape adds no value to colour, but precludes interpretation of the size legend representing number of cells.

2. Results section 2, proliferating myeloid cluster. The authors state that this cluster is 'Specific to the tumor fraction', but this is not consistent with the statistics shown in Fig 2b.

3. Results section 2, Fig. 2j. If the point of this graph is to show that 'CTL-1 abundance was correlated with proliferating T cell cluster, within the tumor fraction', then data points representing normal should not be included (if anything, these appear to suggest a negative correlation), or the colour of the linear regression should be changed to red to avoid confusion.

4. Results section 4, proliferating fibroblasts. These appear to cluster more closely with Pericytes-1 (Fig 4a) and express marker genes for Pericyte-1 (Fig 4c). Please elaborate on how this cluster was labelled.

5. Results section 4, Endo-3 and loss of MEG-3 expression. Fig 4f appears to show that the Endothelial-3 cluster is close to 0 in tumours. Combined with such small numbers, the conclusions regarding the significance of this population in tumour growth in the results section on Endo-3 are a stretch.

6. Results section 4, Fig 4h. Representing differential expression data between only 2 samples, where colour is scaled by row, is a highly misleading way to represent data. This is because regardless of the absolute difference in expression values, the scaling may over-represent the difference by scaling to opposite ends of the colour spectrum. Showing bar plots (as in Ext data 5g) or stacked violins would be recommended.

7. Results, section 5, Fig 5b. How are correlation strengths represented and what are the cut-offs? Please indicate in legend.

8. Results section 5, 'CXCL9/10 signaling via CXCR3 may impact T cell function'. Upregulation on CXCR3 on tumour T cells alone is insufficient to make this claim. If true, what is the difference between the phenotype or transcriptome of CXCR3 expressing and CXCR3 low T cell populations in the data? Similar remarks regarding the CD70-27 axis. Similarly, again, protein level validation is required.

Reviewer #3 (Remarks to the Author): RCC single cell RNA-seq expertise

Alchahin et al describe a new transcriptional metastatic signature that they postulate predicts survival in ccRCC. The study adds unique data, in providing single cell sequencing from two patients with bony metastasis. Analysis of this data provides possible novel insights into RCC biology. The paper is well written and contains interesting discussions. The authors should also be applauded for generating novel data, but there are many points that I would like to see clarified before the broad conclusions can be validated.

Major points:

1) There is no QC data in the entire manuscript. The manuscript states how many cells were sequenced, but I suspect this relates to the number of cells were analysed post QC. There is almost nothing in the methods that would allow a researcher to reproduce the data from the raw sequencing files/ cell ranger output. This is a significant shortfall in research reproducibility.

2) The principal clustering of RCC cells to analyse prognosis and transcriptional pathways uses an algorithm developed to infer copy number aberrations. The application of this method to answer an unrelated question is not justified in this manuscript. Furthermore, the authors are linking these inferred clusters back to the known copy number profiles from the literature. Unfortunately, inferCNV is neither sensitive nor specific to CNV profile, as demonstrated by C2 which shows no CNVs (and the constant looking loss of chr21). We know that ccRCCs all tend to have ubiquitous arm or whole chromosome 3 loss, so it is likely that either the method is not sensitive, or those cells are low quality (as discussed above – the readership has no way of checking the latter possibility as there is no information on either the methods or the results of data QC). In summary, the method is neither sensitive for detecting copy number variants nor there is evidence provided that it is appropriate for clustering cells by transcriptional program. If there is interest in copy number aberrations then DNA sequencing ought to be performed.

3) Sequencing metastases along with primary tumours is of great interest. It would be helpful for the authors to comment on the clinical scenarios that have arisen whereby treatment naïve RCCs have been excised alongside bony metastases. The CARMENA trial demonstrated that upfront cytoreductive nephrectomy should no longer be the default, except for high performance status and low metastatic burden patients. The two metastatic patients described had disease at multiple sites. The clinical readership may require some reassurance that the appropriate clinical course was taken with the above evidence.

4) The cell-cell interaction analysis focusses on two potential axes. The authors infer that these pathways were found in an unbiased manner (ABSTRACT: "An in silico cell-to-cell interaction analysis highlighted"; Later "Several ligands were specifically upregulated in the tumor cell population when compared to adjacent kidney, including SPP1 and CD70"). Where were these genes pathways as compared all others using differential expression? Can you provide the full list? Were they chosen as the top hits/ most significantly enriched, in which case please provide the statistics, or were they chosen as the authors felt they might be interesting?

5) The authors state that "multiple tumors were profiled for some patients". However, when looking at Table S1, it is apparent this equates to single tumours being sampled in different locations, rather than there being multiple primary tumours. This important difference needs clarification.

6) The use of RNA velocity to infer which cell subtype is proliferating is highly error prone. The transcriptional profile of proliferating cells will be dominated by cycling genes and therefore the cluster can easily represent a mix of cell subtypes. One was to infer their origin is to regress out the cycling genes to try to reveal the true nature of the cells through either re-clustering or correlation. Alternatively if one could lineage trace through for instance TCR sequencing, then the origin would be much clearer.

7) Although it is not possible to determine cell sequencing quality through directly looking UMAPs. The authors have not provided any metrics for any readers to determine this. However, particularly the UMAPs for the pericyte clusters look diffuse and possibly low quality.

8) The authors refer to a fibroblast cluster having "the highest EMT gene signature (Fig. 4d), further supporting their role in the promotion of RCC progression". Can this be clarified? The fibroblasts are at no risk of metastasis and therefore an "EMT signature" may not be relevant. Is there a particular ligand/ chemokine that you are talking about that might be promoting EMT in RCC cells?

9) Conclusion (1): It is difficult to argue against that you have generated a signature that predicts survival. However, many such signatures exist, and one needs to know why this signature holds value over the others. Are your methods better or is the dataset more enlightening, such that previous investigators have been unable to derive this? I am not convinced of the above points.

10) Conclusion (2): This is confirmatory. A subset of PT cells have previously been postulated as the origin of ccRCC. No effort have been made however, to show whether the results are in specific agreement or disagreement with the previously highlighted cells of origin.

11) Conclusion (3): It would be valuable to confirm some of the proposed interactions via some of the readily available spatial transcriptomic methods.

RESPONSE TO REVIEWER COMMENTS

Reviewer #1 (Remarks to the Author): expert in clear cell RCC

Alchanin et perform scRNA analysis on a cohort of RCC patients to develop gene base biomarkers. The study adds to the growing list of studies in this space and identifies several possible therapeutic targets. Unfortunately, the paper also suffers from several critical limitations:

Methodological:

Unfortunately, there's no mention at all about handling the possible batch effect issue in this study. At least they should demonstrate there's no/minimal concern of batch effect in sample pooling, for example by PCA analysis. Although they, look to me, mainly used Conos (<https://github.com/kharchenkolab/conos>) for cell clustering and sub-clustering analysis which seems to have measures in integrating data from different samples for clustering. Nevertheless, there's tons of results after clustering from like, for example DEG, pseudo time, velocity and CNV analyses which could be affected by the batch effect. It's a fundamental question that how they make sure batch is not the source of reported results. For example, when comparing the expression difference between tumor cells and normal cells (or mets vs no-mets) from the same cluster, how did they make sure the difference actually came from batch effect?

Thank you for the critical and valid point. Batch effect is an important factor that affects the results of data analysis. We have therefore several different approaches to reduce the impact of batch effects, including:

1. Single cell data integration: Conos was used to integrate multiple scRNA-seq datasets together. In order to measure the potential batch effect for data integration, we included Extended Data Fig 1a which shows major cell populations of individual patients visualized in common joint UMAP embedding. In addition, we also include a UMAP embedding before batch removal as control. As the figures below show there is no clear batch effect after conos alignment.
2. To avoid batch effects on DEG, we perform pseudo-bulk DE analysis for the comparison between sample fractions (for example Tumor Treg vs. Normal Treg). Each sample will form a batch which will limit the potential bias. Recently Jordan et.al benchmarked different single cell DE methods and demonstrated Pseudobulk methods outperform generic and specialized single-cell DE methods¹.
3. For gene expression distance, Expression differences between matching samples were determined by "pseudo-bulk" RNA-seq measurements. We are trying to reduce the bias from patient variability by subsampling the cells by repeating the expression distance procedure 100 sampling rounds. Median expression distance of 100 sampling rounds was used to avoid potential bias. Expression distance was implemented by cacao (<https://github.com/kharchenkolab/cacao>).

In the updated manuscript, we added a section to describe how the batch effect was controlled (See method section, page).

Figure A & B: Conos joint embedding of global landscape of ccRCC before (A) and after (B) control for the batch effect

For CN analysis, since the read count in a sliding window basis of all the “tumor” cells from all the samples were compared to normal nephron cells, again, from all the samples, how can we be sure the normal nephron reference cells are not biased by some samples in read count? And vice versa for the tumor cell read count.

We thank the reviewer for this comment. InferCNV have been widely used to explore tumor single cell RNA-Seq data to identify evidence for large-scale chromosomal copy number variations such as primary glioblastoma², metastatic melanoma³ and metastatic head and neck cancer⁴. To avoid the potential biases from some samples, we take normal nephron cells from multiple patients as reference control. In addition, we evaluated the tumor cell marker genes expression (Extended Data Fig 4d) for malignant cell identification. In the revised manuscript, we also showed metastasis signature gene expression (Fig. 3f) in the individual patient, confirming the expression in the ccRCC metastatic samples instead of the primary ccRCC patients (Fig. 3f).

For velocity analysis, the splicing event classification I believe is individual cell basis so it’s not affected by the batch effect. However, when they calculated the velocity in a group of cells, they seemed to do it on both merged datasets as well as individual sample fraction but it’s not clear how the bath effect was handled in analyzing merged datasets. In the velocity pseudo time analysis, since it’s based on selected number of genes which often came from some kind of DEG which again could be biased by batch effect.

For RNA velocity, we agree with the reviewer that there is a potential batch effect for merged datasets, unfortunately there is no efficient way to remove batch effects, so we created a velocity plot for each individual patient which recapitulates the result from merged datasets. In addition, different data are analyzed under the same parameters and there is no artificial adjustment of DEG. Therefore, the velocity results obtained are meaningful, comparable, and repeatable. In the updated manuscript, we regressed out the cycling genes using seurat and re-clustering the CTLs and proliferating cells. The velocity shows consistent transitions from proliferating T cell population to CTL-1 in merged datasets and individual samples (Fig. 2j and Extended data 3e).

Another point regarding the batch effect is about signature score calculation in scRNA seq. Although it’s mentioned in method that they calculated signature score for each cell as an

average normalized (for cell size) gene expression magnitudes, it's not clear to me what measures were taken in avoiding score inflation or deflation by batch?

We thank the reviewer for pointing this out. We agree with the reviewer regarding the potential batch effects and impact of low cell numbers in some samples/clusters. To ensure robustness of the results and avoid the potential bias, we measure the signature score on a sample level instead of single cell level. In this way, each sample represents a batch, and the impact of cell size is controlled to attain a robust result. Besides the signature score, we also evaluate the key signature gene expression in different cell subpopulations and sample fractions. Macro-1 and Macro-2 resemble M2 phenotype with high expression of CD163, CD68, TGFB1⁵⁻⁷, CCL18⁸, MMP14⁹, TGFB2¹⁰, CTSD¹¹, and MARCO¹² (Extended data 2a-c). In contrast, Macro-3 show high expression of proinflammatory gene expression such as IL1A and IL1B that is associated with M1 phenotype¹³. To illustrate the increased exhaustion signature score in CTL-1, we provided a heatmap to demonstrate exhaustion marker gene expression in each patient (Extended data 3 c-d). As for angiogenesis and EMT, we highlight the enhanced expression of KCNE3, EDNRB, CD36 and INSR (Figure 4j and Extended data 6f). INSR was reported as a marker of the tumor vasculature and can promote tumor angiogenesis¹⁴, while CD36 can promotes the epithelial–mesenchymal transition and metastasis in cancer^{15,16}.

Regarding the survival analysis results, there are a number of questions. First, the p values they reported are actually one-side log-rank test p value as mentioned in the method. When there's no reasonable prior assumption of the direction of the survival difference between groups, 2-side p value should be reported. For example, for the C4 signature (fig 3) survival analysis were they really testing the assumption that C4 high has worse survival as compared to low? Same questions goes to CAF, TREM2 and CTL scores. They only include the top and bottom 20% classified as high and low groups. In this case, to me this undermines the significance level and the robustness of the real biological association between the signature and the survival. They used KIRC and Braun et al cohorts for survival analysis but it's not clear which survival (OS or PFS) and arm were used in the analysis. It will be more informative to see the survival analysis results being tested in the other IO cohorts as well as recurrence cohorts if any.

We kindly thank the reviewer for this comment and agree that 2-side p value should be reported. We therefore used 2-sided test to assess significance of survival analysis for the revised manuscript. Patients were stratified into two groups based on the average expression (binary: top 25% versus bottom 25%) of signature gene derived from scRNA-seq data. In the updated manuscript, we further analyzed signature gene expression within tumor microenvironment, requiring those genes highly expressed in corresponding cell type. Moreover, we evaluated the stability of signature genes listed by bootstrap resampling procedure with p-values at the 95% reproducibility power (see method section). Hereby, we show reliability of the gene signatures and are therefore confident about the survival output. The adjustments for the tumor metastasis signature (Fig. 3g) and CAF (Fig. 4i) survival plots were replaced in the revised manuscript, and remains significant.

As for survival analysis of TREM2+ macrophages, we show TREM2 expression in all immune and non-immune populations through computational analysis, and found that TREM2 is exclusively expressed in the macrophage population, mostly in the Macro-2 subpopulation (Extended data 2f), which suggest that TREM2 is a good marker for tumor

associated macrophages. We also utilized the ccRCC spatial data that was recently published¹⁷ and found infiltrated TREM2+ cells within the ccRCC tumor microenvironment.

As for CTL scores survival analysis, we evaluated the signature gene expression in CTL-1 and rerun the survival with permutation. It did not show significance. Therefore, we withdraw this claim for the association with survival for CTL-1 (Result section 2, last sentence of paragraph 4, manuscript page 8).

Survival curves represent patient OS states, we updated figure legend with detailed description for the survival analysis. As for other IO and recurrence cohorts, unfortunately we did not find any recurrence cohorts and large IO cohorts. The same survival analysis was applied to one additional IO cohorts from Miao et.al (32 patients)¹⁸. We observe the trend of tumor metastasis signature with poor patient overall survival, but no significance due to small sample size.

Clinical:

Table 1 is missing several clinical features such as tumor grade, size gender etc. PRCC case should be excluded as its genomically and immunologically completely distinct. ITH analysis is not addressing Turajlic point of inter-tumor heterogeneity, its more about across tumor heterogeneity. This analysis should be removed.

The authors collected bone mets on two of the patients but do not show any analysis comparing primary vs mets or any bone tropic signatures.

We thank the reviewer for pointing out the unclear details. In supplementary Table 1 we have now included all patient samples and their respective grade/stage. In the table we had mentioned that the pRCC patient was excluded from the analysis. In the revised manuscript we have made it clearer for the reader that the pRCC patient was excluded from all analysis (Manuscript page 5 first paragraph in the result section):

“To provide an overview of the molecular and cellular landscape of patient-matched normal kidney and ccRCC tissue, we performed scRNA-seq profiling (10x Chromium) from freshly resected primary ccRCC tumors (16) and adjacent normal samples from 10 patients (Fig. 1a). Nine patients were diagnosed with ccRCC and 1 patient with papillary RCC, pRCC (pRCC was excluded from analysis)”.

As for bone mets samples, we analyzed the difference between primary ccRCC and metastatic ccRCC (tissue from local kidney metastasis, but diagnosed with bone metastasis). In revised manuscript, we validate the metastatic signature gene expression in tumor cells from bone metastasis sites using an independent study of the bone marrow renal cell carcinoma (RCC) metastasis. Our analysis of seven RCC metastasis cases shows the upregulation of metastatic signature compared to primary tumor cells, further demonstrating the reliability of tumor metastatic signature.

To further substantiate our findings, we performed comparative analysis between ccRCC bone metastatic TME and local kidney ccRCC TME from our other study entitled “Landscape of the human renal cell carcinoma bone metastatic microenvironment”. Here, we collected matched sets of tissue fractions and performed single-cell transcriptomic profiling of cells in solid metastatic tissue (Tumor fraction), liquid bone marrow at the vertebral level of spinal cord compression (Involved), as well as liquid bone marrow from a different

vertebral body distant from the tumor site but within the surgical field (Distal). In addition, we collected publicly available Bone marrow single cell data from Healthy Donors (Healthy) and patients undergoing hip replacement surgery (Benign) as control. Interestingly, we identified a tumor-associated macrophage (TAM) population that was specifically enriched in the patient Tumor fraction samples (see below, Fig. A). Macrophages are heterogeneous and several populations of macrophages have been described in primary ccRCC. We also performed integrated analysis of myeloid cells from primary ccRCC patients. Compared to samples from primary ccRCC, the composition of macrophages in bone metastases demonstrated a shift towards an increased fraction of Macro 2 (see below, Fig. B-D). Macro-2 showed high expression of SPP1, CXCL5, CCL2, CCL7, CCL18 (see below, Fig. C, E). SPP1 is involved in bone formation and in anchoring of osteoclasts to the bone remodeling matrix¹⁹. CCL18 is reported to promote metastasis in breast cancer, colon cancer, and squamous cell carcinoma^{20,21}. CXCL5 is elevated in tumor tissues and is positively associated with lymphatic metastasis and tumor differentiation²².

Figure: **A.** UMAP embedding demonstrating myeloid cell subpopulations from different bone marrow (BM) conditions. **B.** UMAP joint embedding showing integrated analysis of myeloid cells from ccRCC bone metastasis and primary ccRCC. **C.** UMAP embedding showing representative marker gene expression for macrophage subpopulations. **D.** Boxplot comparing proportion of macrophage populations between bone metastatic ccRCC and primary ccRCC. Statistical significance were accessed by Wilcox run sum test. **E.** Dot plots showing cytokine gene expression across different myeloid subsets. The color represents scaled average expression of marker genes in each cell type, and the size indicates the proportion of cells expressing marker genes.

Reviewer #2 (Remarks to the Author): expert in immunogenomics

Summary

In this study, Alchahin et al. analyse single-cell RNA transcriptomes of samples derived 10 treatment naïve renal cell carcinoma (RCC) patients, including biopsies from the tumour core, metastases, and adjacent normal tissue, generating data on 35,000 normal kidney cells and 122,000 tumour cells. They begin by identifying several immune and stromal cell types that are differentially abundant in tumour versus adjacent normal tissue, and transcriptional state differences within these cell types, which are pronounced in myeloid cells, cytotoxic cells, fibroblasts, endothelial cells, and pericytes. These observations set the stage for subsequent sub-cluster analysis. Alchahin et al. describe a population of TREM2+ SPP1+ macrophages that may associate with outcome and identify an ‘exhausted’ cytotoxic T lymphocyte (CTL) cluster which associates with tumour Treg activity. By inferring copy-number variation from single cell transcriptomes, the authors identify a cluster of cells which appear in patients who develop metastases. Differentially expressed genes in this cluster were labelled as a ‘metastatic signature’ which associates with poor survival and disease stage. Within stromal cells, endothelial cells and pericytes from the tumour fraction were enriched in genes associated with angiogenesis, indicating a role in vascular remodelling. Similarly, cancer associated fibroblasts (CAFs) highly expressed genes related to epithelial-mesenchymal transition (EMT), which relate to survival. Finally, from ligand-receptor pair analysis, the authors identify CD70-CD27 and CXCL9/10-CXCR3 signalling axes that may be important in the tumour microenvironment.

Overall, there are strengths in the sampling strategy that includes both adjacent normal kidney and some metastases (although the latter is limited in number). However, while there is adequate quantitative analysis characterising the transcriptional differences between tumour and normal tissue, there are several short-comings which limit enthusiasm, principally a failure to validate single cell RNA-seq findings at a protein level, but also there are some analytical weaknesses, a rudimentary description of methods and this work does recapitulate previous single cell studies in terms of inferring a proximal tubular origin of RCC.

Major comments:

1. Results fig 1 – pericyte cluster is very substantial, even in normal tissue. Are the authors sure these are pericytes? What is the expression of canonical pericyte markers such as CD31, NG2, PDGFR beta, CD146, Nestin? RGS5 expression is not specific to pericytes, similarly MYH11 is expressed in other cell types, such as smooth muscle cells.

Thank you for the important remarks giving us a chance to clarify our approach regarding the stroma population. These are important remarks since we show that the pericyte population significantly increases in tumor compared to normal, and that the merged endothelial population is significantly decreased in tumor compared to normal. Previous studies performing single-cell RNA-sequencing analysis defined stromal populations^{23,24} expressing for instance RGS5. However, we carefully explored the reviewers' comments and therefore first explored the expression pattern of NG2 (CSPG4), PDGFRB and MCAM (CD146) suggested pericyte marker genes from the reviewer, and we show that the expression is evident and are predominantly in the tumor compared to normal in our dataset (Extended data 7e).

Figure: UMAP embedding of stromal cells showing expression of pericyte markers.

2. Results fig 1 – in a similar vein, the marker genes stated for monocytes are non-specific. Both FCN1 and S100A8/9 are highly expressed in neutrophils. Additional information is needed on the expression of monocyte and neutrophil-specific genes – e.g, FCGR3B for neutrophils.

We understand the reviewer’s point and therefore re-run our analysis. As the reviewer mentioned S100A8/9 are highly expressed in neutrophils, as well as serving a role in inflammation and immune regulatory function in myeloid cells²⁵. When expression plots were made (shown below) for commonly known neutrophil markers as FCGR3B (suggested from reviewer), AZU1²⁶, LCN2²⁷ and ELANE²⁸, we could not detect these genes in our myeloid populations. Therefore, we could further strengthen our conclusive annotations made for the monocytes, and that we have no neutrophils in our dataset.

Figure: UMAP embedding of myeloid population showing no expression of common neutrophil markers.

3. Results fig 1 – proportional changes in cell types. The conclusions about relative enrichment of different cell types is made based on representation in scRNAseq data. This is problematic, as different cell types are preferentially preserved during the generation of 10x data, and this is not uniform between tissues, for example, a more hypoxic tissue like a tumour will see more cell attrition during processing and loading. Any statements on abundance of cell types between tumour and normal needs to be validated by an alternative modality, e.g. flow cytometry.

We kindly thank the reviewer for this comment. We understand the concerns regarding proportional changes in cell types generated from single cell data analysis, and have addressed this point by validating populations of interest through flow cytometry. We performed analysis on biobanked patient material of newly diagnosed ccRCC patients. We therefore chose to validate with flow cytometry some key populations to convince the reviewers that our data is robust:

- TREM2+ Macro-2 population that is significantly increased in the tumor compared to normal (Extended data 2g, i).
- In the stromal population the endothelial population is significantly decreased in tumor compared to normal (Extended Data 1c, 7d-e).
- CD27 expression on CD8+, PD-1+ show a significant increase in tumor compared to normal, and CD27 expression show a positive trend towards an increase in tumor compared to normal (Fig. 5g; Extended data 8e-g).

To further substantiate our data, we further utilize spatial data from a recently published study¹⁷ to demonstrate that there are TREM2+ macrophages and T cells infiltrating the ccRCC tumor. Here, for the stroma populations we also observe expression of genes from the different subsets in the tumor, for instance in pericyte-1, which we see is significantly increased in tumor compared to normal (Fig. 4e). We show that the markers PDGFRb, RGS5 and THY1 that are mainly expressed in pericyte-1 are also expressed in 2 different ccRCC patients (Extended data 7a-c). These are mainly expressed in the subpopulation Pericyte-1, which we show is significantly increased in tumor compared to the normal (Fig. 4e). When we investigate markers expressed by Pericyte-2, which is significantly decreased in tumor compared to normal (Fig. 4e), we observe the same trend in the spatial data when looking at the expression of CNN1 and RERGL (Extended data 7a-c), where they are not expressed to the same extent as the genes of Pericyte-1, further supporting our single-cell data.

4. Results 2, referencing fig 2b. see comment 4 re: making conclusions about whether a cell type is less abundant in tumour. This would need to be confirmed by an alternative method like FACS or imaging. Furthermore, the comparing the reduction of tumour-resident CD1C+ mDCs to circulating populations is not necessarily relevant as the source they may arise from pre-cursors or monocytes rather than enter the tumour as differentiated cDCs. It should also be made clear that reference 27 is a study on blood immune populations in breast cancer, not RCC.

We have taken the reviewer's perspective into consideration and a claim may be rough in regard to the reduction of the CD1C+ mDC population in tumors compared to healthy individuals. Hence, we decided to remove the statement and reference 27 for this part:

” ...consistent with studies showing that frequencies of circulating DCs are significantly lower in cancer patients compared with healthy individuals”.
(See result section 2, first paragraph page 6).

5. Results 2, macrophage populations. The fact that there is differential abundance of Macro-2 in tumours, which have a higher M2 score is not sufficient to merit the claim that they are ‘suppressive and support tumour growth’. Are there any gene expression patterns to support a directly suppressive role for these cells, eg, ARG1, PDL1/2? Additionally, expression of CD163 and CSF1R (which the authors identify as M2 marker genes) are similar in Macro-2 and Macro-3. Furthermore, CSF1R is not an M2s-specific macrophage gene.

We agree with the reviewer and have performed additional analysis. Our new analysis reveals three different macrophage subpopulations and suggest that Macro-1 and Macro-2 resembles the M2 macrophages compared to Macro-3. When we investigate signature scores, we take into consideration different genes and their co-expression that is discussed in literature. We addressed this point by adding more markers for macrophages with M2 phenotype that are recognized in literature such as *CD163*, *CD68*, *TGFBI*⁵⁻⁷, *CCL18*⁸, *MMP14*⁹, *TGFB2*¹⁰, *CTSD*¹¹, and *MARCO*¹²(Extended data 2a-d). Compared to Macro-1 and Macro-2 that express these markers, Macro-3 show high expression of proinflammatory gene expression such as *IL1A* and *IL1B* that is associated with M1 phenotype¹³.

6. Results 2, macrophage populations. M1 versus M2 gene signature – it is not evident (I can’t find this info in the methods) what genes were included in this M1 and M2 scores. Were these gene sets derived from mouse or human macrophages?

We thank the reviewer for emphasizing this issue. For the macrophage population we annotated them based on collective genes from literature on human data. The scores were generated by the co-existence of defined genes for M1 and M2. The gene list can be found in Supplementary Table 5, which is based on published data using scRNA-seq data²⁹.

7. Results 2, SPP1+ TREM2+ Macro-2 population. Firstly, validation of the presence of the TREM2 macrophage population in tumour v normal is required. Secondly, if validated that TREM2-expressing macrophages are enriched in the tumour fraction, scoring TREM-2 gene expression in bulk RNAseq data (Ext Data 2g) and concluding that TREM2+ macrophages play a role in ccRCC progression is problematic. The authors do not show that the differential expression of TREM2+ in the bulk RNAseq can be solely attributed to macrophages. We suggest that, at a minimum, the authors show data on TREM2 expression in other tumour cell populations (both immune and non-immune), and perform deconvolution of the bulk data, to make claims on the role of this population on RCC outcome. Moreover, this statistical approach is not robust. If one were to take 30,000 genes and scored each one of these in a Kaplan-Meier analysis with LogRank p-value cut-off of 0.05, then over 1000 genes would come up as significant for association with survival. In addition, for all KM analyses, the definitions of ‘high’ and ‘low’ for TCGA/CheckMate cohort stratification should be more clearly defined.

We appreciate the reviewers' criticism and have taken this into consideration and performed validation analysis. As reviewer suggested, we now show TREM2 expression in all immune and non-immune populations through computational analysis and found that TREM2 is exclusively expressed in the macrophage population, mostly in the Macro-2 subpopulation (Extended data 2a, f), suggesting TREM2 is a good marker for Macro-2. We further validated

TREM2 on protein level with flow cytometric analysis comparing seven matched normal adjacent kidney tissue and ccRCC tumors. We found a significant increase when comparing the adjacent normal kidney tissue with the tumor. Here, we see an overall increasing trend in TREM2 expression in the ccRCC tumors among the patients (Extended data 2g, i). Hence supporting our conclusive findings regarding TREM2 expressing Macro-2. As for the robustness of survival analysis, Benjamini-Hochberg (BH) method was used to adjust p-value across all genes. Our results retain significance for TREM2.

Finally, to further validate our findings, we utilized the ccRCC spatial data that was recently published¹⁷ and found infiltrated TREM2+ cells within the ccRCC tumor microenvironment (Extended data 2k).

8. Results 2, Ext data 3a. The plot inserted here is incorrect and is a repeat of Fig 2b.

We thank the reviewer for their thorough read. We have adjusted this (Extended Data 3a) where we added the fraction of T cells.

9. Results 2, CTL-1 exhaustion score. What does this CTL-1 exhaustion score consist of? reference 33 is a systematic review with no specific reference to the gene set used for scoring. If there is an overlap between genes used to score exhaustion and Treg activity (which is also not described), then the analysis in Fig 2h and Ext. data 3f would be fundamentally flawed. Moreover, the authors do not indicate how CTL-1 exhaustion is scored, and if this were done by simple module scoring or gene set enrichment, then it is incorrect to conclude that this CTL-1 population associates with survival (Ext. data 3g), rather it is the genes in this gene set that do.

We thank the reviewer for this comment. We generated the scores based on published literature. In the included supplementary Table 5 we have listed the specific genes and the references. We further added an expression plot with recognized exhaustion markers as PD-1, TIGIT, LAG3, TIM3 and CTLA4³⁰⁻³² (Extended data 3b, c) to substantiate our exhaustion score signature. Besides signature score, we provided a heatmap to demonstrate exhaustion marker gene expression in each patient (Extended data 3c-d).

Regarding the correlation between exhaustion and Treg activity score, we first evaluated the correlation in single cell data where signature score was computed using different cell types (Treg activity in Tregs and exhaustion signature score CTL-1). There may be a bias in bulk RNA-seq data due to five overlapping genes. In the updated manuscript, we removed the overlapped genes and recalculated the signature score. As Extended data 3f shows, we still observe a significant correlation between exhaustion and Treg activity score in complementary to cell abundance correlation.

As for the survival analysis, we evaluated the signature gene expression in CTL-1 and rerun the survival with a bootstrap resampling procedure of signature genes. It did not show significance. Therefore, we withdraw this claim for the association with survival for CTL-1 (Result section 2, last sentence of paragraph 4, manuscript page 8).

10. Results section 2, Fig 2i. Similar comment to 6). How was 'Treg activity' scored? Without showing that genes in this signature are not differentially expressed outside the Treg compartment or without performing deconvolution, the effect on survival cannot be attributed to Tregs.

Similar to the previous comment, in supplementary Table 5 we have now included all the genes that we used to define the different populations. Based on published data we could identify them as active Tregs due to their expression of CD25 (IL2RA), FOXP3 and CTLA-4^{6,33,34} (Fig.2i). As mentioned in the previous comment from the reviewer, we rerun the survival with a bootstrap resampling procedure of signature genes. It did not show significance and therefore withdraw the significance of Treg association with poorer survival (Result section 2, last sentence of paragraph 4, manuscript page 8).

11. Results section 2, RNA velocity analysis. The use of RNA velocity analysis to infer differentiation trajectories in the current format is problematic for several reasons.

1) The fact that CTL-1 and proliferating T cells are differentially abundant in the tumour is not sufficient grounds to isolate these 2 clusters only for RNA velocity analysis. If so, there are only 2 possibilities for the root/parent cluster identified, as is inferred by the authors, which is not how RNA velocity should be used.

2) Further, if it is true that the red cluster represents proliferating T cells and the yellow cluster exhausted T cells, given that RNA velocity measures unspliced-to-spliced transcripts, then this will likely bias the analysis to select proliferating cells as the root, given they are more transcriptionally active. Indeed, RNA velocity analysis on data where there are clusters with significantly higher levels of cell proliferation often results in large arrows originating from the proliferating cluster.

3) Visualisation – the RNA velocity analysis should not be performed on a sub-setted graph derived from Fig 2d. A new graph and new embeddings should be computed for CTL-1 and proliferating T cells, with RNA-velocity overlaid. We suggest that subset selection be expanded to all T cells, or CTLs only. In the latter, it may be necessary to show that the proliferating T cells are related to CTLs only. It may be worth considering whether these are stem-like (TCF1+/TCF7+) T cells. Importantly, the conclusion that proliferating T cells give rise to the exhausted CTL-1 population must be toned down.

We thank the reviewer for these valid points. Firstly, as the reviewer suggested, we regressed out cell cycle genes and generated a new graph and a new embedding for the CTL and proliferating T cells. We found that CTL-1, CTL-2 and the proliferating T cells cluster show similar patterns in the RNA velocity as well (Fig. 2j-k; Extended Data 3e) where the proliferating T cells appear to differentiate into CTL-1.

We further addressed the points regarding the TCF1/TCF7 expression as mentioned by the reviewer to see if proliferating T cells are stem-cell like. However, we found that TCF1+/TCF7+ are not highly expressed in the proliferating T cells, instead, as we show below, we observe exhaustion signature genes (PDCD1/TOX) being expressed in the proliferating T cells suggesting proliferating phenotype of exhausted T cells.

Figure A & B: UMAP embedding of exhausted markers TOX and PDCD1 compared to the requested marker TCF7 (A). UMAP embedding showing exhausted signature score expression pattern (B).

12. Results section 3, inferred CNV profiles and metastatic signature. There is insufficient evidence here to argue that genes associated with cluster 4 represent a metastatic signature. 3 out of 4 primary tumour samples which had corresponding metastases, and metastasis samples from a single patient had cells belonging to this cluster. The cell number from each of these samples is less than 250. While, there may be an association here, the claim that differential genes of this cluster represent ‘a distinct signature predictive of metastatic potential’, based solely on qualitative observations, is overreaching.

Thank you for this point. Indeed, there is huge variation of number of tumor cells. In order to increase the reliability of our data, we also include three ccRCC metastases patients (P55, P913, P916) from public data³⁵ into our analysis. In the revised manuscript, we evaluated metastatic signature gene expression in the individual patient, showing expression of the metastatic signature genes in the ccRCC metastatic samples instead of the primary ccRCC patients, as we demonstrate in the figure below. In addition, we also validate the metastatic

Figure: Metastatic signature gene expression pattern visualized in violin plot comparing primary ccRCC, local kidney metastasis and bone metastasis.

signature gene expression in tumor cells from bone metastasis sites using an independent study of human RCC bone metastasis. Our analysis of seven RCC bone metastasis cases shows the upregulation of metastatic signature compared to primary tumor cells, further demonstrating the reliability of tumor metastatic signature (Figure 3f). We also updated the method section with a detailed description of how to define metastatic signatures.

13. Results section 3, C4 signature. The authors show that 3 out of 4 genes in their ‘C4 signature’ are more highly expressed in late-stage disease (Fig 3h, Ext. data 4f). They then show that this same signature also associates with poor survival (Fig 3g). This means that the survival analysis in 3g may simply reflect disease stage, which is the definition of poor outcome. Hence, the claim made in the manuscript’s conclusion, stating ‘We identify a distinct metastatic signature that predicts survival outcome’, while not necessarily incorrect, has important caveats to disclose, in conjunction with point 12).

We thank the reviewer for this comment. We are emphasizing the defined signature genes because they are highly expressed in tumor cells and further upregulated in the metastatic tumor cells (see below, Fig. A). Therefore, they could act as a metastatic signature. Similar to point 12, in the revised manuscript, analysis of seven RCC metastasis cases shows the upregulation of metastatic signature compared to primary tumor cells, further demonstrated the predictivity of tumor metastasis.

As for high expression in late-stage disease, according to the disease stages of ccRCC, stage 4 is considered as metastatic and that the tumor has reached distant sites³⁶. We think that the results further support our conclusion on the metastatic signature. The survival analysis in Fig. 3g may not be stable, and simply reflects disease stage. To address this issue, we first updated survival analysis and evaluated the stability of signature genes listed by bootstrap resampling procedure with p-values at the 95% reproducibility power (see method section). Hereby, we show reliability of the gene signatures and are therefore confident about the survival output. The adjustments for the metastatic signature (Fig. 3g) survival plots were replaced in the revised manuscript, and it remains a significant result. In addition, we also want to highlight that the patients in the CheckMate cohort are mostly in late stage. Moreover, we perform the same survival analysis on stage 4 patients only. We observe the trend of tumor metastasis signature with poor overall patient survival, but no significance due to small sample size (see below, Fig. B).

Lastly, we addressed this comment by first rephrasing the second sentence in the Discussion (Discussion section, first paragraph, page 14 to:

“...We uncovered intratumor heterogeneity, as well as a highly heterogeneous tumor microenvironment, and multiple immunosuppressive cell-cell interactions”.

Figure A & B: Metastatic signature genes visualized in violin plot in C4 (A). Survival probability of C4 (B).

14. Results section 4, stromal cell populations. In this section, several claims are made regarding very rare cell populations (eg. Endothelial-3 or Fibroblasts proliferating). The authors should demonstrate that more than a single patient / sample contributes to these rare populations to prove that these are not merely a sequencing batch effect or poor integration. If this is true, then they need to validate the presence of these cells using an alternative technology to single cell RNAseq e.g, spatial method.

We thank the reviewer for this comment. We have now generated Supplementary Table 3 for cell frequency of all patients and the stroma populations, as well as the cell numbers comparing normal with tumor. In this table we confirm that endothelial-3 present in multiple patients.

Regarding the proliferating fibroblasts, we re-annotated the proliferating fibroblasts to proliferating stroma cells since the population expresses ACTA2 and lacks the common fibroblast markers as DCN and LUM (see results section 4, last sentence in the third paragraph, page 11 and supplementary Table 3). Since it is a small population, we did not make further claims on this population.

Viewing the endothelial population, we find that it is significantly decreased in tumor compared to normal (Extended data 1c) and which we validated by flow cytometry and further confirmed that endothelial cells are significantly decreased in tumor compared to normal (Extended data 7d).

Analyzing the different subsets of the stromal cells in the tumor, focusing on the pericytes, we found that pericyte-1 is significantly increased in tumor, whereas pericyte-2 show an opposite phenotype (Fig. 4e). We utilized public spatial data recently published¹⁷ to try and distinguish these subsets of pericytes pericyte markers and validate our findings. Here, we show in two patients that pericyte markers are solely expressed in pericyte-1, including PDGFRb, RGS5 and THY1 (Extended data 7a-c).

When we investigate markers expressed by Pericyte-2, which is significantly decreased in tumor compared to normal (Fig. 4e), we can see a similar trend in the spatial data when investigating expression of CNN1 and RERGL that are expressed in pericyte-2 (Fig. 4c). These two genes do not appear to be expressed in the spatial data of ccRCC tumors (Extended data 7a-c) further supporting our single-cell data.

15. Results section 4 – claims like ‘a notable proportion of the pericytes, and some of the endothelial and the fibroblast cells were only detected in the tumor fraction (Fig. 4a, b),’ need to be validated, confirming the presence of the cell types claimed to be tumour-unique or enriched in tumour and absence in normal, for example by using flow cytometry employing marker genes for these clusters identified by sc RNAseq, or by imaging, eg, RNAscope or immunofluorescence microscopy.

Thank you for the important remarks giving us a chance to clarify our claims and findings. These are important remarks since we show that the pericyte population significantly increased in tumor compared to normal. We have referenced studies where they performed single-cell RNA-seq analysis and defined stromal populations^{23,24}. Furthermore, we explore the expression pattern of known pericyte markers shown below.

Moreover, we put a genuine effort to quantify the stroma populations, including the pericytes in the ccRCC tumors and their matched normal tissue. Due to the time constraint, we were only able to use frozen single cell suspensions of some of the patients saved from the single cell experiments. With the material we had available, we tried to quantify the pericytes at the protein level with flow cytometry using NG2 (Supplementary Table 8. Catalog nr: MA5-28549, Thermofischer). Unfortunately, we were not able to quantify the pericytes and we believe that it may be due to the frozen sample and that pericytes might be sensitive to freezing. On the other hand, we were able to quantify the endothelial population and show that they significantly decreased in tumor compared to normal (Extended data 7d) as shown in Extended Data 1c.

Next, we sought to validate the shift in cell abundance in the endothelial and pericytes subset. We utilize spatial transcriptomics data¹⁷ and evaluate representative marker gene expression of the stromal cell subsets. Endo-1 (PLVAP and CA2) was found to be significantly enriched in the tumor compartment compared to normal, whereas, Endo-3 abundance (EHD3, EMCN and PI16) significantly decreased in tumor (Fig. 4c, f). In a spatial context, the tumor infiltrated region was defined by CA9 expression. We found high expression of PLVAP, CA2 and low/no expression of EHD3, EMCN, PI16 expression in tumor infiltrated region,

which supports our single cell data that Endo-1 is increased while Endo-3 is decreased in the tumor microenvironment (Extended data 7a-c).

Similarly, we explored the pericyte subpopulations to validate our findings in Fig. 4e. We show that gene markers, PDGFRb, RGS5, THY1 that are expressed in pericyte-1 (see below, Fig. B), the subpopulation that is significantly increased in tumor compared to normal (Fig. 4e), are also expressed in the spatial data (Extended data 7a-c). Again, when investigating the markers expressed by Pericyte-2, which is significantly decreased in tumor compared to normal (Fig. 4e), we can see a similar trend in the spatial data looking at the expression of CNN1 and RERGL that are expressed in pericyte-2 (Fig. 4c). These two genes do not appear to be expressed in the spatial data of ccRCC tumors (Extended data 7a-c) further supporting our single-cell analysis (Extended data 7d).

16. Results section 4, ‘tumor fibroblasts exhibit the highest EMT gene signature, further supporting their role in the promotion of RCC progression and metastasis’. The latter part of this statement is not supported by the data shown in Fig 4d where the EMT signature is not significantly different in tumour or normal fibroblasts.

Thank you for this remark. We have re-run the analysis for significance of the EMT of tumor fibroblasts. We observe that there is no significance due to high variation of sample size. We therefore withdraw this sentence:

“...Consistent with this, tumor fibroblasts exhibited the highest EMT gene signature (Fig. 4d), further supporting their role in the promotion of RCC progression and metastasis”.

(Result section 4, paragraph 3, page 11).

17. Results section 4, Fig 4g. While GO terms may be useful in reducing dimensions when considering large numbers of genes, they tend to be unspecific, especially given that there are 7500 GO BP terms, which may erroneously detect significant pathways. In addition to this figure, it would be more convincing to demonstrate the genes driving this association, for example, by identifying key genes driving the association in endothelial cells, or gene-set enrichment analysis in a ‘pseudo-bulk’ manner.

We are grateful to the reviewer for this comment. We performed Pseudo-bulk DE analysis for the comparison between sample fractions (for example Tumor Fibroblast vs. Normal Fibroblast). Recently Jordan *et.al* benchmarked different single cell DE methods and demonstrated Pseudobulk methods outperform generic and specialized single-cell DE methods¹. To explore the genes driving this association, we provided another supplementary table that shows differential expressed genes between tumor and normal for each stroma cell subtypes (Supplementary Table 6). We also labeled differential expressed cytokines, transcription factor, kinase and chromatin regulators. Interestingly, we observed increased expression of Insulin Receptor (INSR) in tumor endo-1 (shown in Fig. A below). INSR was reported as a marker of the tumor vasculature that can promote tumor angiogenesis (PMID: 30559346). This is consistent with the GO enrichment analysis showing top upregulated pathways in blood vessel development and angiogenesis (shown in Fig. B below). Besides GO enrichment analysis, we also run gene-set enrichment analysis using clusterprofiler³⁷, showing similar enrichment result.

Figure A & B: UMAP embedding showing expression of INSR (A). GO enrichment analysis of top upregulated pathways in subpopulation Endothelial-1 (B).

18. Results section 4, Endo-2 Fig 4g, ‘Top upregulated pathways for Endo-2 include protein localization to the ER, which could indicate an increase in demand for protein synthesis and secretion’. Following on from point 18), protein trafficking and processing genes are pathways that appear repeatedly when applying GO term analysis to multiple data sets in a highly nonspecific manner. Thus, the conclusion regarding Endo-2 here based solely on a single GO term is speculation at best.

We thank the reviewer for their constructive criticism. The sentence: “Top upregulated pathways for Endo-2 include protein localization to the ER, which could indicate an increase in demand for protein synthesis and secretion” is speculative and was therefore removed (page 12). The sentence that Endo-2 may have a supportive role for RCC progression based on reference 62 and 63 was also removed (Result section 4, last sentence of paragraph 5, page 12).

19. Results section 4, Fig 4i. Similar comments on previous KM analyses. What are the genes used for scoring? What are the cut-offs used for cohort stratification? The analysis only shows that the genes are relevant rather than CAFs.

We kindly thank the reviewer for this comment and agree that the analysis may indicate the genes rather than CAFs. To address this issue, we have tested two different deconvolution methods, however these did not appear to distinguish relatively similar subpopulations such

as (Macro-1, Macro-2 and Macro-3, fibroblast and CAFs). Instead, in the revised manuscript, we analyzed signature gene expression within tumor microenvironment, requiring those genes highly expressed in corresponding cell type. Moreover, we evaluated the stability of signature genes listed by bootstrap resampling procedure with p-values at the 95% reproducibility power (see method section). Hereby, we show reliability of the gene signatures and are therefore confident about the survival output. The adjustments for the CAF (Fig. 4i) survival plots were replaced in the revised manuscript and it remains a significant result

20. Results, endothelial cell clusters. Moreover, the 3 paragraphs on Endothelial cell clusters 1-5, spanning over 1.5 pages, is highly descriptive and simply restates what is already shown in figure 4 and Ext. data 5, and is speculative.

We thank the reviewer for this comment. We have addressed the reviewer comment by reducing repetitive or over speculative statements. See modified manuscript version of the result section.

We removed text from the result section 4, page 11 according to below:

- “Consistent with this, tumor fibroblasts exhibited the highest EMT gene signature (Fig. 4d), further supporting their role in the promotion of RCC progression and metastasis.”
- “In line with those findings, tumor Endo-1 displayed a high angiogenesis score (Fig. 4d) (VWF, HSPG2, SPRY1) and expressed the key receptors KDR and FLT-1 (also known as VEGFR-1 and VEGFR-2, Fig. 4c, Extended data Fig. 5b, d)⁶¹. The top upregulated pathways of Endo-1 were those involved in blood vessel and circulatory system development, indicating increased angiogenesis, further supporting the annotation of Endo-1 as TECs (Fig. 4g).”
- “Top upregulated pathways for Endo-2 include protein localization to the endoplasmic reticulum (ER), which could indicate an increase in the demand for protein synthesis and secretion (Fig. 4g).”

21. Results section 5 – predicted interactions - to conclude that CD70 represents a therapeutic target in RCC the authors at least need to validate expression of CD70 and CD27 at a protein level on the cell types they claim are of relevance – eg – protein level expression of CD27 on tumor-associated CTL-1, proliferating T-cells and Tregs.

This is a critical point! We thank the reviewer for highlighting this. To our knowledge, CD70 is commonly known to be expressed on ccRCC tumors^{38,39} and others are currently investigating CD70 as a treatment target expressed on ccRCC with potential to metastasize⁴⁰⁻⁴². In ccRCC spatial data we validate that CD70 is expressed in tumor cells (CA9+ and CD70+ cells)¹⁷ (Fig. 5h).

We next focused on investigating the receptor of CD70, CD27, that is known to be expressed on cytotoxic T cells and regulatory T cells⁴³. Our computational analysis shows that CD27 is significantly increased in the tumor associated CTL-1 compared to the normal (Fig. 5d). Moreover, we also utilized the spatial data and validated the CD27 expression in exhausted T cells (CD8+, PDCD1+, LAG+ cells)¹⁷ (Extended data 9d).

We further tried to validate CD27 by flow cytometry on exhausted cytotoxic T cells and regulatory T cells. We explored the co-existence of CD27 with PD-1 in CD8+ ccRCC tumor cells compared to the normal adjacent kidney tissue, and the CD27 with LAG3 in CD8+ tumor cells. In addition, we tried to validate CD27 on Tregs by investigating expression of CD4+, CD25+, FOXP3+ on T cells. Regarding expression of on the T cell population we demonstrate that CD8+, PDCD1+, CD27+ cell population is significantly increased in 4 patients with ccRCC compared to their adjacent normal tissue taken from the same patient (Fig. 5g). For CD27 expression in the CD8+, LAG3+ T cells (in the same patients) we do see a positive trend, though not significant, similar to the previous statement (Extended data 8g).

For the expression of CD27 on Tregs, we detect a small population (flow analysis data), however due to the low cell count number, we are careful to make further conclusions regarding the expression of CD27 on Treg. We conclude that the lack of significant difference in the Tregs may be impacted by patient heterogeneity and that more patient numbers need to be added in order to demonstrate significance. Unfortunately, we were limited with access to adding more patients during the revision period. However, combining our single cell data, and the significance of the expression of CD27 in CD8+, PDCD1+ cell population and the correlation between CD27 and PDCD1 from a public dataset (Extended data 9c), we believe that our data show reliable results. Lastly, we want to conclude that CD27 is yet a possible target in ccRCC, specifically since its ligand, expressed on tumors, has already been suggested as a target by others^{38,40,41}, where we focus more on the expression of its receptor and the cell populations that expresses it in ccRCC. Using the spatial public dataset¹⁷ of ccRCC tumor we find CD27 expression with TME, and also co-localized with CD70 (Extended data 9d).

A

B

Figure: Gating strategy for Tregs (A). Statistical analysis of 6 matched patients, paired t-test, $p=0,9367$ (B).

We have now based on the new data modified the sentence “The CD27 receptor was overexpressed in tumor-associated CTL-1, proliferating T-cells and Tregs” in the manuscript on page 13, result section 5, second paragraph to:

“The CD27 receptor was overexpressed in tumor-associated CTL-1”.

22. Results section 5, figure 5f. From this data, 1 of 3 conclusions can be drawn: 1) CD27 expression is associated with CTL-1 exhaustion; 2) CD70 expression is associated with CTL-1 exhaustion; 3) CD27 and CD70 in combination is associated with CTL-1 exhaustion. To make the conclusions that the authors are claiming, i.e., the CD70-CD27 interaction is important in the TME, 1) and 2) have to be ruled out.

We thank the reviewer for this comment. To our knowledge, CD70 has already been well studied in ccRCC and has been concluded in studies to be expressed on the tumor of ccRCC hence being a reasonable target for this disease on the tumor^{38,39}. For suggested conclusion #2, we calculated the correlation between CD70 expression (tumor cells) and CTL-1 exhaustion score, and did not find a significant association (Extended data 9a). However, its receptor CD27⁴³ is significantly associated with CTL-1 exhaustion (Extended data 9a). As mentioned in the response to the reviewer's previous comment, we manage to show that there is a significantly increased CD27 expression in CD8+PDCD1+ (Fig. 5g) and a positive trend towards an increase of CD27 expression in CD8+LAG3+ T cells (Extended data 8g) in the ccRCC compared to its adjacent normal kidney tissue. We aim to highlight the importance of the receptor CD27 in ccRCC, to eventually hinder tumor progression.

We further use bulk RNA sequencing data to verify this association, and found that CD27 significantly correlates with PDCD1/LAG3 expression (Extended data 9c).

23. Results section 5, figure 5j. It is unclear what this figure is trying to show, and the legend associated is not informative and may potentially have an error?

We thank the reviewer for their comment. Figure 5j is taken from a Tumor Immune Dysfunction and Exclusion database^{44,45}, showing the expression correlation between PDCD1, CXCL9, CXCL10, CD70, CD27, CXCR3 and CTLs. We found these genes being significantly correlated, further implying the immune suppression function. In the revised manuscript, we updated the figure legend and moved this figure to supplementary figures.

24. Discussion, ‘unexpected intratumor heterogeneity’. The reviewers do not believe that the manuscript has uncovered ‘unexpected’ heterogeneity, as it is largely as previously described in scRNAseq datasets.

We kindly thank the reviewer for pointing this out to help us improve our manuscript. We adjusted by removing the word “unexpected”: “We uncovered intratumor heterogeneity, as well as a highly heterogeneous tumor microenvironment, and multiple immunosuppressive cell-cell interactions” (see first paragraph in the discussion, page 14).

25. Discussion, ‘an increase in Tregs expressing HAVCR2 is accompanied by increase of exhaustion of CTL-1 in the tumor fraction. This particular immunosuppressive cell population of Tregs may represent another potential therapeutic target in ccRCC.’

The authors suggest that these Tregs represent an additional therapeutic target independent of the PD-1 and CTLA-4 axis. But, are these TIM-3+ Tregs different from the PD-1 and CTLA-4 expressing Tregs?

Thank you for the important point. This is indeed not supposed to be misinterpreted. We removed the sentence since that is not what we are addressing (see page 15 in the manuscript).

Minor comments:

1. Introduction, paragraph 3 – typo ‘(several?)’ appears to be an edit that should be deleted.

Again, we thank the reviewer for their critical eye. This has been corrected in the revised version of the manuscript.

2. Introduction, final paragraph. ‘...tumor cells are transcriptionally similar to a subset of proximal tubule cells which may be an indication of the tumor cell of origin²³,’ – in correct reference – The correct reference here is Young, Mitchell, Braga et al. Science (2019) (reference 25). This paper presented one of the earliest and largest scRNAseq studies in RCC, and is the key publication to cite when describing transcriptional similarity between the RCC tumour cell of origin and PT, and when citing RCC scRNAseq papers. This reference should similarly be included in later results and discussion sections discussing this point.

The reference is adjusted, see reference #17 in the revised manuscript, page 4.

1. Figure 1h. This MDS plot highlighting different expression distances between tumour and normal samples adds little value and is unsurprising. Moreover, visualizing tumour and normal by shape adds no value to color, but precludes interpretation of the size legend representing number of cells.

We thank the reviewer for their help of visualization of the data. We addressed this comment by keeping the shapes and changed the color of normal to red (Fig. 1h).

2. Results section 2, proliferating myeloid cluster. The authors state that this cluster is ‘Specific to the tumor fraction’, but this is not consistent with the statistics shown in Fig 2b.

Thank you for pointing this out. We show that this population exist more in the tumor, however the population may not be tumor specific. Hence, we rephrased the sentence:

- “We identify an increasing population of proliferating myeloid clusters expressing ...” (see manuscript page. 6, first sentence of result section 2).’

3. Results section 2, Fig. 2j. If the point of this graph is to show that ‘CTL-1 abundance was correlated with proliferating T cell cluster, within the tumor fraction’, then data points representing normal should not be included (if anything, these appear to suggest a negative correlation), or the color of the linear regression should be changed to red to avoid confusion.

We have addressed this comment by removing the normal samples from the dot plot (Fig. 2j).

4. Results section 4, proliferating fibroblasts. These appear to cluster more closely with Pericytes-1 (Fig 4a) and express marker genes for Pericyte-1 (Fig 4c). Please elaborate on how this cluster was labelled.

We addressed this comment by double checking the annotation and decided to call them “proliferating stroma cells” instead of “proliferating fibroblasts” due to the low/lack of expression of the common fibroblast markers DCN (lack of) and LUM (low expression). Half of the island shows an expression of, for instance, ACTA2 (Fig. 4a, c).

5. Results section 4, Endo-3 and loss of MEG-3 expression. Fig 4f appears to show that the Endothelial-3 cluster is close to 0 in tumours. Combined with such small numbers, the conclusions regarding the significance of this population in tumour growth in the results section on Endo-3 are a stretch.

We thank the reviewer for this comment. We added this to emphasize the phenotype of endo-3 compared to the other endothelial groups. We understand that it is speculative due to the low numbers, hence we shortened the section. See modified text for Endo-3 in result section 4, page 12.

The Endothelial-3 population is significantly lower in the tumor compared to their adjacent normal kidney tissue (Fig. 4f). We observe this population in multiple patients, however in a variety of cell numbers which could be due to inter-patient differences and the tumor profile (Supplementary Table 3). Details of removed text from result section 4 regarding Endo-3, see manuscript page 12:

- “.....by preventing the binding of leukocytes to the endothelium, reduce tumor proliferation and increase tumor cell apoptosis”” The loss of MEG3-expressing Endo-3 GECs might therefore benefit tumor growth by dysregulating inflammatory and immune binding processes”.

6. Results section 4, Fig 4h. Representing differential expression data between only 2 samples, where color is scaled by row, is a highly misleading way to represent data. This is because regardless of the absolute difference in expression values, the scaling may over-represent the difference by scaling to opposite ends of the colour spectrum. Showing bar plots (as in Ext data 5g) or stacked violins would be recommended.

We thank the reviewer for highlighting the visual context of this figure. We addressed this comment by creating a violin plot (Fig. 4h).

7. Results, section 5, Fig 5b. How are correlation strengths represented and what are the cut-offs? Please indicate in legend.

Again, we are grateful for the time put to comment on the minor errors. We expand method section with detail description of how to filter significant ligand-receptor pairs. We also update figure legend and include another supplementary Table 7 for all significant ligand-receptor pairs (Fig. 5b; Extended data 8h).

8. Results section 5, ‘CXCL9/10 signaling via CXCR3 may impact T cell function’. Upregulation on CXCR3 on tumour T cells alone is insufficient to make this claim. If true, what is the difference between the phenotype or transcriptome of CXCR3 expressing and

CXCR3 low T cell populations in the data? Similar remarks regarding the CD70-27 axis. Similarly, again, protein level validation is required.

Thank you for this comment. Indeed, we cannot argue for the impact on T cell function. However, we show that CXCL9/10 are significantly increased in tumor compared to normal (Fig. 5i) and has been associated with metastasis⁴⁶. We computationally demonstrate that CXCL9/10 are overexpressed by the myeloid markers and the CXCR3 by the T cells in the tumor (Extended data 9e). We want to hypothesize that this interaction may impact the microenvironment and potentially promote tumor progression through deregulation of inflammatory pathways. We find that both CTL-1 and Treg in the tumor express CXCR3 significantly more than the adjacent normal (see figure below and Extended data 9e). We modified the last sentence of result section 5 (same sentence copied by the reviewer), page 13:

We therefore hypothesize that CXCL9/10 signaling via CXCR3 may impact the microenvironment and potentially promote tumor progression through deregulation of inflammatory pathways.

Figure: CXCR expression in Treg (A-B). Expression of exhausted marker HAVCR2 in Treg (C).

Reviewer #3 (Remarks to the Author): RCC single cell RNA-seq expertise

Alchahin et al describe a new transcriptional metastatic signature that they postulate predicts survival in ccRCC. The study adds unique data, in providing single cell sequencing from two patients with bony metastasis. Analysis of this data provides possible novel insights into RCC biology. The paper is well written and contains interesting discussions. The authors should also be applauded for generating novel data, but there are many points that I would like to see clarified before the broad conclusions can be validated.

Major points

1) There is no QC data in the entire manuscript. The manuscript states how many cells were sequenced, but I suspect this relates to the number of cells were analysed post QC. There is almost nothing in the methods that would allow a researcher to reproduce the data from the raw sequencing files/ cell ranger output. This is a significant shortfall in research reproducibility.

We thank the reviewer for this important comment. In the revised manuscript, we have updated the method section with a detailed description of data quality control. We also include a Supplementary Table 3 for cell QC and cell frequency. In addition, we created a github page (<https://github.com/shenglinmei/ccRCC.analysis/>) which enables the reproducibility of the analysis.

2) The principal clustering of RCC cells to analyse prognosis and transcriptional pathways use an algorithm developed to infer copy number aberrations. The application of this method to answer an unrelated question is not justified in this manuscript. Furthermore, the authors are linking these inferred clusters back to the known copy number profiles from the literature. Unfortunately, inferCNV is neither sensitive nor specific to CNV profile, as demonstrated by C2 which shows no CNVs (and the constant looking loss of chr21). We know that ccRCCs all tend to have ubiquitous arm or whole chromosome 3 loss, so it is likely that either the method is not sensitive, or those cells are low quality (as discussed above – the readership has no way of checking the latter possibility as there is no information on either the methods or the results of data QC). In summary, the method is neither sensitive for detecting copy number variants nor there is evidence provided that it is appropriate for clustering cells by transcriptional program. If there is interest in copy number aberrations then DNA sequencing ought to be performed.

Thank you for this point. We addressed this comment by first highlighting that our goal is to identify the signature genes that can predict tumor metastasis instead of copy number aberrations. Even though InferCNV have been widely used to explore tumor single cell RNA-Seq data to identify evidence for large-scale chromosomal copy number variations such as primary glioblastoma², metastatic melanoma³ and metastatic head and neck cancer⁴, it still has limitation for CNV calls and clonal evolution. To demonstrate the metastatic signatures, in revised manuscript, we validate the metastatic signature gene expression in tumor cells from bone metastasis sites using an independent study of the bone marrow renal cell carcinoma (RCC) metastasis. Our analysis of seven RCC metastasis cases shows the upregulation of metastatic signature compared to primary tumor cells, further verified the predictivity of tumor metastasis (see figure below and Fig. 3f). In addition, we also updated survival analysis to evaluate the stability of signature genes listed by bootstrap resampling procedure with p-values at the 95% reproducibility power (see method section). Hereby, we

show reliability of the gene signatures and are therefore confident about the survival output. The adjustments for the metastatic signature survival plots (Fig. 3g) were replaced in the revised manuscript, and it remains a significant result.

As for C2, we do see weak signals of CNV and did not observe a clear difference of data quality of C2 cluster. In the revised manuscript we rephased the sentence:

“C2 did not show CNV aberration, but demonstrated an alerted transcriptional profile (VEGFA, NNMT and NDUFA4L2) as compared to normal kidney (Fig. 3f).”

We also updated the method section with a detailed description of data quality control in Supplementary table 3.

We agree with the reviewer that single cell DNA sequencing could help to address the tumor cell subclones, however, due to time limitations and not the scope of this study, we therefore kindly ask to retract the need for DNA sequencing.

A

B

Figure: Metastatic signature gene expression pattern visualized in violin plot comparing primary ccRCC, local kidney metastasis and bone metastasis (A). Survival probability of C4 in two independent cohort studies (B).

3) Sequencing metastases along with primary tumors is of great interest. It would be helpful for the authors to comment on the clinical scenarios that have arisen whereby treatment naïve RCCs have been excised alongside bony metastases. The CARMENA trial demonstrated that upfront cytoreductive nephrectomy should no longer be the default, except for high performance status and low metastatic burden patients. The two metastatic patients described had disease at multiple sites. The clinical readership may require some reassurance that the appropriate clinical course was taken with the above evidence.

We thank the reviewer for this highly clinically relevant question. We report data from two cases relevant to this concern (i.e. two patients who had spine decompression + resection of the primary tumor). The surgeries for these patients were in 2017 (06/09/2017 spine surgery + 06/15/2017 nephrectomy for the first patient; 10/27/2017 spine surgery + 12/04/2017 nephrectomy for the second patient). The CARMENA trial was ongoing from 2009 until 2017, with its final analysis in October 2018. The conclusions were published in 2018⁴⁷, 2019⁴⁸ and 2021⁴⁹. You are correct that CARMENA results have changed practice such that these two patients would likely have been managed differently (i.e., systemic therapy without nephrectomy subsequent to spinal cord decompression) if they had presented after the CARMENA results were published. As informed by the CARMENA data, we had no further decompression/nephrectomy cases in 2018 or beyond.

4) The cell-cell interaction analysis focusses on two potential axes. The authors infer that these pathways were found in an unbiased manner (ABSTRACT: “An *in silico* cell-to-cell interaction analysis highlighted”; Later “Several ligands were specifically upregulated in the tumor cell population when compared to adjacent kidney, including SPP1 and CD70”). Where were these gene pathways as compared to all others using differential expression? Can you provide the full list? Were they chosen as the top hits/ most significantly enriched, in which case please provide the statistics, or were they chosen as the authors felt they might be interesting?

We much appreciate the reviewer pointing this out. Besides the permutation test, we also use additional computational steps to filter ligand and receptor pairs including cell-type specific expression and DEG analysis comparing tumor vs adjacent normal. In the filtering step, we first filter significant ligand-receptor pairs based on permutation test (adjust-p value 0.05), then evaluate if ligand or receptor is upregulated in corresponding cell type comparing tumor vs adjacent normal. In addition, we measured both ligand and receptor expression levels across cell types, requiring both ligand and receptor highly expressed in corresponding cell types taking other cell types as background.

In the revised manuscript, we updated the method section and provided the full ligand-receptor list in the Supplementary Table 7.

5) The authors state that “multiple tumors were profiled for some patients”. However, when looking at Table S1, it is apparent this equates to single tumours being sampled in different locations, rather than there being multiple primary tumours. This important difference needs clarification.

We rephrased the sentence to highlight that “for two patients, tumor from different locations from the kidney were profiled”. (See first result section, first paragraph, manuscript page 5)-of the Rephrase: “From some patients, tumor from different locations were profiled”.

6) The use of RNA velocity to infer which cell subtype is proliferating is highly error prone. The transcriptional profile of proliferating cells will be dominated by cycling genes and therefore the cluster can easily represent a mix of cell subtypes. One way to infer their origin is to regress out the cycling genes to try to reveal the true nature of the cells through either re-clustering or correlation. Alternatively, if one could trace lineage through for instance TCR sequencing, then the origin would be much clearer.

We highly appreciate the reviewer's comment. As the reviewer suggested, we regressed out cell cycle genes and generated a new graph and a new embedding for the CTLs and proliferating T cells. We found that CTL-1, CTL-2 and the proliferating T cells cluster show similar patterns in the RNA velocity as well (Fig. 2k, Extended Data 3e) where the proliferating T cells appear to differentiate into CTL-1.

As for TCR sequencing, we agree with the reviewer that TCR sequencing could help to address the clone expansion of exhausted T cell, however, due to time limitations and not the scope of this study, we therefore kindly ask to retract the need for TCR sequencing.

7) Although it is not possible to determine cell sequencing quality through directly looking UMAPs. The authors have not provided any metrics for any readers to determine this. However, particularly the UMAPs for the pericyte clusters look diffuse and possibly low quality.

We appreciate the reviewer's concern, we have provided an additional supplementary Table 3, which contains detailed QC information for each cell. In addition, we also provided another figure showing the distribution of total UMI of each cell type (see figure below). We therefore reassure that the data and the quality of the data appear to be reliable. In addition, we provide another table of frequency of each cell population in each patient (Supplementary Table 3).

Figure: Quality of data for the stromal populations

8) The authors refer to a fibroblast cluster having “the highest EMT gene signature (Fig. 4d), further supporting their role in the promotion of RCC progression”. Can this be clarified? The fibroblasts are at no risk of metastasis and therefore an “EMT signature” may not be

relevant. Is there a particular ligand/ chemokine that you are talking about that might be promoting EMT in RCC cells?

Thank you for this remark. We re-analyzed the significance of the EMT of tumor fibroblasts. We observe that there is no significance due to high variation of sample size. We therefore withdraw this sentence from result section 4, middle of paragraph 3, manuscript page 11):

- ... “Consistent with this, tumor fibroblasts exhibited the highest EMT gene signature (Fig. 4d), further supporting their role in the promotion of RCC progression and metastasis”.

9) Conclusion (1): It is difficult to argue against that you have generated a signature that predicts survival. However, many such signatures exist, and one needs to know why this signature holds value over the others. Are your methods better or is the dataset more enlightening, such that previous investigators have been unable to derive this? I am not convinced of the above points.

Previous studies have mainly performed on bulk RNA sequencing and have had limitations to identify genes in single cell level. In recent years the emergent technologies have enabled the field to investigate more in depth into the cancer microenvironment. The advantage of single-cell RNA-sequencing approaches is that it enables us to yield a higher resolution and in-depth analysis of a high range of cells. Researchers within the field of RCC have used this approach previously^{6,35,50}. However, all groups have focused on different aspects. Until today, there has not been an in-depth investigation at a single cell level of the gene signatures that, when co-expressed, are associated with metastasis. This has been claimed for each of the genes in separate studies. Compared to existing data, the purpose of the project is to compare the microenvironment of the adjacent normal kidney tissue and tumor from the same patient to further understand what may be influencing the disease progression. We highly believe that these biological insights will help the field of urological oncology, biology, genetics and eventually the clinical aspects to tackle this disease. And hopefully at an earlier stage.

We collected the public prognostic signature genes⁵¹⁻⁵³, and further evaluated the expression levels of those prognostic genes within ccRCC tumor TME. We found that those signature genes are expressed in multiple cell types (see figure below). However, our proposed genes such as the metastatic signature in tumor, TREM2 in macrophage population (Extended data 2e) and the CAF population in the stroma appear to be expressed in a specific cell population (Fig. 4h-i).

Figure: Dotplot showing gene expressions in all cell types.

10) Conclusion (2): This is confirmatory. A subset of PT cells has previously been postulated as the origin of ccRCC. No effort has been made however, to show whether the results are in specific agreement or disagreement with the previously highlighted cells of origin.

We thank the reviewer for this important comment. Young et al. found a subset of PT cells could be the origin of ccRCC⁵⁰. We further exploited the fact that they were defined by SLC17A3 and VCAM1 with absence of SLC7A13 within our data. Their data explanation for the origin of ccRCC aligns with our annotated proximal tubule 2 (PT2) and PT1. When investigating marker genes, we observed an upregulation of SOX9, IL32 and SLC22A6 in our data which was confirmed with public datasets when comparing normal with tumor. However, we find that SLC22A8 and SLC22A6, in our PT1, significantly decreased in tumor compared to normal (Extended data 4c, d). Therefore, our final hypothesis is that PT2 may be the origin population of ccRCC.

A**B**
Figure: UMAP embedding genes expressed in the proximal tubule (A). Boxplot of genes that are increased and decreased in the tumor (B).

11) Conclusion (3): It would be valuable to confirm some of the proposed interactions via some of the readily available spatial transcriptomic methods.

We kindly thank the reviewer for this comment. In the revised manuscript, we utilized publicly available spatial transcriptomic to support our finding¹⁷. Regarding the CD70-CD27 interaction that we want to highlight that we put some extra effort to validate our computational findings.

To our knowledge, CD70 is commonly known to be expressed on ccRCC tumors^{38,39} and research groups are currently working on establishing targeted treatment against CD70 expressing ccRCC with potential to metastasize⁴⁰⁻⁴². In ccRCC spatial data we also observed that CD70 is expressed in tumor cells (CA9+ and CD70+ cells), as well as CD27 expression¹⁷ (Extended data 9d).

We next focused on emphasizing its receptor CD27 that is known to be expressed on cytotoxic T cells⁴³. Our computational analysis shows that CD27 is significantly increased in the tumor associated CTL-1 compared to the normal (Fig. 5d). Moreover, we also utilized the spatial data and verified the CD27 expression in CD8+ T cells¹⁷ (Extended data 9d).

We further attempted to validate CD27 by flow cytometry on exhausted cytotoxic T cells and regulatory T cells. We explored the co-existence of CD27 with PD-1 in CD8+ ccRCC tumor cells compared to the normal adjacent kidney tissue, and the CD27 expression on LAG3 in CD8+ tumor cells. We demonstrate that CD8+, PDCD1+, CD27+ cell population is significantly increased in four patients with ccRCC compared to their adjacent normal tissue taken from the same patients (Fig. 5g). For CD27 expression in the CD8+, LAG3+ T cells we do see a positive trend, though not significant, similar to the what we see in the CD27 expression in CD8+ and PDCD1+ T cells (Extended data 8g).

We conclude that the significance may be impacted by patient heterogeneity and that more patient numbers need to be added in order to show significance. However, combining our single cell data, and the significance of the expression of CD27 in CD8+, PDCD1+ cell population and the correlation between CD27 and PDCD1 from a public dataset (Extended data 9c), we believe that our data show reliable results. Lastly, we want to conclude that CD27 is yet a possible target in ccRCC, specifically since its ligand, expressed on tumors, has already been suggested as a target^{38,40,41}, where we focus more on the expression of its receptor and the cell populations that expresses it in ccRCC.

References

1. Squair, J.W., *et al.* Confronting false discoveries in single-cell differential expression. *Nat Commun* **12**, 5692 (2021).
2. Patel, A.P., *et al.* Single-cell RNA-seq highlights intratumoral heterogeneity in primary glioblastoma. *Science* **344**, 1396-1401 (2014).
3. Tirosh, I., *et al.* Dissecting the multicellular ecosystem of metastatic melanoma by single-cell RNA-seq. *Science* **352**, 189-196 (2016).
4. Puram, S.V., *et al.* Single-Cell Transcriptomic Analysis of Primary and Metastatic Tumor Ecosystems in Head and Neck Cancer. *Cell* **171**, 1611-1624 e1624 (2017).
5. Chung, W., *et al.* Single-cell RNA-seq enables comprehensive tumour and immune cell profiling in primary breast cancer. *Nat Commun* **8**, 15081 (2017).
6. Braun, D.A., *et al.* Progressive immune dysfunction with advancing disease stage in renal cell carcinoma. *Cancer Cell* (2021).
7. Obradovic, A., *et al.* Single-cell protein activity analysis identifies recurrence-associated renal tumor macrophages. *Cell* **184**, 2988-3005.e2916 (2021).
8. Wu, Y., *et al.* Spatiotemporal Immune Landscape of Colorectal Cancer Liver Metastasis at Single-Cell Level. *Cancer Discov* **12**, 134-153 (2022).
9. Liang, Z.W., *et al.* M2-phenotype tumour-associated macrophages upregulate the expression of prognostic predictors MMP14 and INHBA in pancreatic cancer. *J Cell Mol Med* **26**, 1540-1555 (2022).
10. Wolf-Dennen, K., Gordon, N. & Kleinerman, E.S. Exosomal communication by metastatic osteosarcoma cells modulates alveolar macrophages to an M2 tumor-promoting phenotype and inhibits tumoricidal functions. *Oncoimmunology* **9**, 1747677 (2020).
11. Xu, B., *et al.* Establishment and Validation of a Genetic Label Associated With M2 Macrophage Infiltration to Predict Survival in Patients With Colon Cancer and to Assist in Immunotherapy. *Front Genet* **12**, 726387 (2021).
12. Shi, B., *et al.* The Scavenger Receptor MARCO Expressed by Tumor-Associated Macrophages Are Highly Associated With Poor Pancreatic Cancer Prognosis. *Front Oncol* **11**, 771488 (2021).

13. Jumeau, C., *et al.* Expression of SAA1, SAA2 and SAA4 genes in human primary monocytes and monocyte-derived macrophages. *PLoS One* **14**, e0217005 (2019).
14. Nowak-Sliwinska, P., *et al.* Oncofoetal insulin receptor isoform A marks the tumour endothelium; an underestimated pathway during tumour angiogenesis and angiostatic treatment. *Br J Cancer* **120**, 218-228 (2019).
15. Deng, M., *et al.* CD36 promotes the epithelial-mesenchymal transition and metastasis in cervical cancer by interacting with TGF- β . *J Transl Med* **17**, 352 (2019).
16. Nath, A., Li, I., Roberts, L.R. & Chan, C. Elevated free fatty acid uptake via CD36 promotes epithelial-mesenchymal transition in hepatocellular carcinoma. *Sci Rep* **5**, 14752 (2015).
17. Meylan, M., *et al.* Tertiary lymphoid structures generate and propagate anti-tumor antibody-producing plasma cells in renal cell cancer. *Immunity* **55**, 527-541.e525 (2022).
18. Miao, D., *et al.* Genomic correlates of response to immune checkpoint therapies in clear cell renal cell carcinoma. *Science* **359**, 801-806 (2018).
19. Nollet, M., *et al.* Autophagy in osteoblasts is involved in mineralization and bone homeostasis. *Autophagy* **10**, 1965-1977 (2014).
20. Chen, J., *et al.* CCL18 from tumor-associated macrophages promotes breast cancer metastasis via PITPNM3. *Cancer Cell* **19**, 541-555 (2011).
21. Korbecki, J., Olbromski, M. & Dziegiel, P. CCL18 in the Progression of Cancer. *Int J Mol Sci* **21**(2020).
22. Mao, Z., *et al.* CXCL5 promotes gastric cancer metastasis by inducing epithelial-mesenchymal transition and activating neutrophils. *Oncogenesis* **9**, 63 (2020).
23. Hartmann, D.A., *et al.* Pericyte structure and distribution in the cerebral cortex revealed by high-resolution imaging of transgenic mice. *Neurophotonics* **2**, 041402 (2015).
24. Bondjers, C., *et al.* Transcription profiling of platelet-derived growth factor-B-deficient mouse embryos identifies RGS5 as a novel marker for pericytes and vascular smooth muscle cells. *Am J Pathol* **162**, 721-729 (2003).
25. Yang, J., *et al.* Calcium-Binding Proteins S100A8 and S100A9: Investigation of Their Immune Regulatory Effect in Myeloid Cells. *Int J Mol Sci* **19**(2018).
26. Naito, T., Jingushi, K., Ueda, K. & Tsujikawa, K. Azurocidin is loaded into small extracellular vesicles via its N-linked glycosylation and promotes intravasation of renal cell carcinoma cells. *FEBS Lett* **595**, 2522-2532 (2021).
27. Xie, X., *et al.* Single-cell transcriptome profiling reveals neutrophil heterogeneity in homeostasis and infection. *Nat Immunol* **21**, 1119-1133 (2020).
28. Cui, C., *et al.* Neutrophil elastase selectively kills cancer cells and attenuates tumorigenesis. *Cell* **184**, 3163-3177.e3121 (2021).
29. Azizi, E., *et al.* Single-Cell Map of Diverse Immune Phenotypes in the Breast Tumor Microenvironment. *Cell* **174**, 1293-1308.e1236 (2018).
30. Lu, L., Bai, Y. & Wang, Z. Elevated T cell activation score is associated with improved survival of breast cancer. *Breast Cancer Res Treat* **164**, 689-696 (2017).
31. Zheng, C., *et al.* Landscape of Infiltrating T Cells in Liver Cancer Revealed by Single-Cell Sequencing. *Cell* **169**, 1342-1356.e1316 (2017).
32. Wherry, E.J. & Kurachi, M. Molecular and cellular insights into T cell exhaustion. *Nat Rev Immunol* **15**, 486-499 (2015).
33. Delacher, M., *et al.* Single-cell chromatin accessibility landscape identifies tissue repair program in human regulatory T cells. *Immunity* **54**, 702-720.e717 (2021).
34. Chevrier, S., *et al.* An Immune Atlas of Clear Cell Renal Cell Carcinoma. *Cell* **169**, 736-749.e718 (2017).

35. Bi, K., *et al.* Tumor and immune reprogramming during immunotherapy in advanced renal cell carcinoma. *Cancer Cell* (2021).
36. Warren, A.Y. & Harrison, D. WHO/ISUP classification, grading and pathological staging of renal cell carcinoma: standards and controversies. *World J Urol* **36**, 1913-1926 (2018).
37. Wu, T., *et al.* clusterProfiler 4.0: A universal enrichment tool for interpreting omics data. *Innovation (Camb)* **2**, 100141 (2021).
38. Jilaveanu, L.B., *et al.* CD70 expression patterns in renal cell carcinoma. *Hum Pathol* **43**, 1394-1399 (2012).
39. Junker, K., *et al.* CD70: a new tumor specific biomarker for renal cell carcinoma. *J Urol* **173**, 2150-2153 (2005).
40. Ji, F., *et al.* Targeting the DNA damage response enhances CD70 CAR-T cell therapy for renal carcinoma by activating the cGAS-STING pathway. *J Hematol Oncol* **14**, 152 (2021).
41. Pal, S.K., *et al.* A phase 1 trial of SGN-CD70A in patients with CD70-positive, metastatic renal cell carcinoma. *Cancer* **125**, 1124-1132 (2019).
42. Kathuria-Prakash, N., Drolen, C., Hannigan, C.A. & Drakaki, A. Immunotherapy and Metastatic Renal Cell Carcinoma: A Review of New Treatment Approaches. *Life (Basel)* **12**(2021).
43. van de Ven, K. & Borst, J. Targeting the T-cell co-stimulatory CD27/CD70 pathway in cancer immunotherapy: rationale and potential. *Immunotherapy* **7**, 655-667 (2015).
44. Fu, J., *et al.* Large-scale public data reuse to model immunotherapy response and resistance. *Genome Med* **12**, 21 (2020).
45. Jiang, P., *et al.* Signatures of T cell dysfunction and exclusion predict cancer immunotherapy response. *Nat Med* **24**, 1550-1558 (2018).
46. Groom, J.R. & Luster, A.D. CXCR3 ligands: redundant, collaborative and antagonistic functions. *Immunol Cell Biol* **89**, 207-215 (2011).
47. Méjean, A., *et al.* Sunitinib Alone or after Nephrectomy in Metastatic Renal-Cell Carcinoma. *N Engl J Med* **379**, 417-427 (2018).
48. Arora, S., *et al.* Cytoreductive Nephrectomy: Assessing the Generalizability of the CARMENA Trial to Real-world National Cancer Data Base Cases. *Eur Urol* **75**, 352-353 (2019).
49. Méjean, A., *et al.* Sunitinib Alone or After Nephrectomy for Patients with Metastatic Renal Cell Carcinoma: Is There Still a Role for Cytoreductive Nephrectomy? *Eur Urol* **80**, 417-424 (2021).
50. Young, M.D., *et al.* Single-cell transcriptomes from human kidneys reveal the cellular identity of renal tumors. *Science* **361**, 594-599 (2018).
51. Li, P., Ren, H., Zhang, Y. & Zhou, Z. Fifteen-gene expression based model predicts the survival of clear cell renal cell carcinoma. *Medicine (Baltimore)* **97**, e11839 (2018).
52. Büttner, F., *et al.* Clinical utility of the S3-score for molecular prediction of outcome in non-metastatic and metastatic clear cell renal cell carcinoma. *BMC Med* **16**, 108 (2018).
53. Kawashima, A., *et al.* Immunological classification of renal cell carcinoma patients based on phenotypic analysis of immune check-point molecules. *Cancer Immunol Immunother* **67**, 113-125 (2018).

REVIEWERS' COMMENTS

Reviewer #1 (Remarks to the Author):

satisfied with the authors responses to all reviews

Reviewer #2 (Remarks to the Author):

The authors have made a substantial effort to address many of our concerns, including the provision of essential methodological detail and protein level validation of TREM2+ macrophages. These substantially improve the paper and I am satisfied with these responses.

Reviewer #4 (Remarks to the Author):

The authors have addressed most of the comments from Reviewer 3. However, there are still some issues that need to be fixed:

1. Supplementary Table 3 was uploaded in Excel Xls format. Xls only supports a maximum of 65536 rows and 256 columns. Thus, part of the cell metadata was lost since more than 157k cells were generated. It is suggested for the authors re-export the data into Xlsx format, which supports 1048576 rows and 16384 columns.
2. The authors described the quality control steps by removing cells having fewer than 800 total UMI and more than 20% mitochondrial transcripts, and used the Scrublet package to identify doublets. The authors did not set an upper cutoff for the total UMI, and it is clear that some outliers exist from the violin plot in the data quality for the stromal populations. Regarding how doublet identification affects the total UMI after QC, it is suggested to provide more detailed comparisons for pre- and post-QC metrics in the method section. It is also confusing that the authors state that 157,881 cells were sequenced (line 119), and 157,881 were obtained after QC.
3. In Supplementary Table 3, cells with total UMI < 700 and MT.ratio > 20% were still present. Was Supplementary Table 3 supposed to include all post-QC cells? Based on the authors' preprocessing code, the total UMI cutoff seems to be 700 (<https://github.com/shenglinmei/ccRCC.analysis/blob/34e4d9f04822408ddb248b4b3cf0f31d16e2a32b/preprocess.r#L19>). Please clarify the quality control steps.
4. Figure 1h stated that t-SNE embedding was used to project expression distances. However, in line 151, the projection method was mentioned as multidimensional scaling. Please clarify.
5. In the 'METHODOLOGY - RNA velocity-based cell fate tracing' section, the name of the velocity package should be velocityto, and the typo 'UMPA' should be 'UMAP'.
6. The column names did not match their values in Supplemental Table 2.

RESPONSE TO REVIEWER COMMENTS

Reviewer #3:

1. Supplementary Table 3 was uploaded in Excel Xls format. Xls only supports a maximum of 65536 rows and 256 columns. Thus, part of the cell metadata was lost since more than 157k cells were generated. It is suggested for the authors re-export the data into Xlsx format, which supports 1048576 rows and 16384 columns.

We thank the reviewer for this comment and have accordingly replaced Supplementary Table 3 with Xlsx format.

2. The authors described the quality control steps by removing cells having fewer than 800 total UMI and more than 20% mitochondrial transcripts, and used the Scrublet package to identify doublets. The authors did not set an upper cutoff for the total UMI, and it is clear that some outliers exist from the violin plot in the data quality for the stromal populations. Regarding how doublet identification affects the total UMI after QC, it is suggested to provide more detailed comparisons for pre- and post-QC metrics in the method section. It is also confusing that the authors state that 157,881 cells were sequenced (line 119), and 157,881 were obtained after QC.

We apologize for the unclear statement in our manuscript. In total, we sequenced 175,485 cells, and obtained 157,881 cells after quality control. We have revised manuscript and method section with a more detailed comparisons for pre- and post-QC metrics. In addition, we evaluated the distribution of total UMI and only found a small proportion of cells with extreme high total UMI. 130 cells have more than 20,000 total UMI, among them, 30 cells have more than 30,000 total UMI. As reviewer suggested, we also evaluated the relationship between total UMI and doublet identification. Total UMI is significantly increased in doublets, suggesting the potential admixture of cells.

Figure legend: **a.** Histogram showing distribution of total UMI per cell, red line indication cutoff at $\log_{10}(20,000)$ and $\log_{10}(30,000)$. **b.** Comparison of total UMI difference between predicted doublets (scrublet scores above 0.4) and singlets. Statistics significance is accessed by Wilcoxon rank sum test.

3. In Supplementary Table 3, cells with total UMI < 700 and MT.ratio > 20% were still present. Was Supplementary Table 3 supposed to include all post-QC cells? Based on the authors' preprocessing code, the total UMI cutoff seems to be 700 (<https://github.com/shenglinmei/ccRCC.analysis/blob/34e4d9f04822408ddb248b4b3cf0f31d16e2a32b/preprocess.r#L19>). Please clarify the quality control steps.

We apologize for this error. Supplementary Table 3 is supposed to be post-QC cells. We double checked the code, the correct total UMI cut-off is 700 and we didn't find cells with < 700 total UMI presented in Supplementary Table 3. As for MT ratio, we found mitochondrial transcripts ratio filtering step was not successfully applied in initial data analysis, we corrected the statement in METHODOLOGY and further evaluated distribution of mitochondrial transcripts ratio. As Figure below shows, 972 cell have a relative high MT ratio (>0.3) and most of those cells are malignant tumor cells and normal kidney nephron cells. The role of mitochondria in cancer formation and progression has been widely studied, Mitochondria supply energy, control redox homeostasis and oncogenic signaling, which can promote cancer development^{1,2}. It's also not a surprise to see high mitochondrial activity in kidney nephron cells, kidney requires a large number of mitochondria to remove waste and mitochondria provide the energy to drive these important functions³. Based on this, we concluded that high MT cells in our study could be biologically meaningful and do not represent a high fraction of apoptotic or lysing cells.

Figure legend: a. Histogram of mitochondrial transcripts ratio and the red line represent MT ratio=0.3. b. Cell type frequency of high MT ratio cells (>0.3, 972 cells).

4. Figure 1h stated that t-SNE embedding was used to project expression distances. However, in line 151, the projection method was mentioned as multidimensional scaling. Please clarify.

We apologize for this error. Figure 1h supposed to be projection of multidimensional scaling (MDS), we have revised the figure legend.

5. In the 'METHODOLOGY - RNA velocity-based cell fate tracing' section, the name of the velocity package should be velocyto, and the typo 'UMPA' should be 'UMAP'.

We appreciate the suggestions from the reviewer. We have corrected the package name and typo.

6. The column names did not match their values in Supplemental Table 2.

We apologize for the unclear in Supplemental Table 2. We have updated the Supplemental Table 2.

References

1. Zong, W.X., Rabinowitz, J.D. & White, E. Mitochondria and Cancer. *Mol Cell* **61**, 667-676 (2016).
2. Hsu, C.C., Tseng, L.M. & Lee, H.C. Role of mitochondrial dysfunction in cancer progression. *Exp Biol Med (Maywood)* **241**, 1281-1295 (2016).
3. Bhargava, P. & Schnellmann, R.G. Mitochondrial energetics in the kidney. *Nat Rev Nephrol* **13**, 629-646 (2017).